# CHiLS: Zero-shot Image Classification with Hierarchical Label Sets

## Abstract

Open vocabulary models (e.g. CLIP) have shown strong performance on zero-shot classification through their ability generate embeddings for each class based on their (natural language) names. Prior work has focused on improving the accuracy of these models through prompt engineering or by incorporating a small amount of labeled downstream data (via finetuning). In this paper, we aim to tackle classification problems with coarsely-defined class labels. We propose Classification with Hierarchical Label Sets (or CHiLS), an alternative strategy that proceeds in three steps: (i) for each class, produce a set of subclasses, using either existing label hierarchies or by querying GPT-3; (ii) perform the standard zero-shot CLIP procedure as though these subclasses were the labels of interest; (iii) map the predicted subclass back to its parent to produce the final prediction. Across numerous datasets (with implicit semantic hierarchies), CHiLS leads to improved accuracy yielding gains of over 30% in situations where known hierarchies are available and more modest gains when they are not. CHiLS is simple to implement within existing CLIP pipelines and requires no additional training cost.

## 1 Introduction

Recently, machine learning researchers have become captivated by the remarkable capabilities of pretrained *open vocabulary models* (Radford et al., 2021; Wortsman et al., 2021; Jia et al., 2021; Gao et al., 2021; Pham et al., 2021; Cho et al., 2022; Pratt et al., 2022). These models, like CLIP (Radford et al., 2021) and ALIGN (Jia et al., 2021), learn to map images and captions into shared embedding spaces such that images are close in embedding space to their corresponding captions but far from randomly sampled captions. The resulting models can then used to assess the relative compatibility of a given image with an arbitrary set of textual "prompts". Notably, Radford et al. (2021) observed that by inserting each class name directly within a natural language prompt, one can then use CLIP embeddings to assess the compatibility of an images with each among the possible classes. Thus, open vocabulary models are able to perform zero-shot image classification, and do so with high rates of success (Radford et al., 2021; Zhang et al., 2021b).

Despite the documented successes, the current interest in open vocabulary models poses a new question: **How should we represent our classes for a given problem in natural language?** As class names are now part of the inferential pipeline (as opposed to mostly an afterthought in traditional scenarios) for models like CLIP in the zero-shot setting, CLIP's performance is now directly tied to the descriptiveness of the class "prompts" (Santurkar et al., 2022). While many researchers have focused on improving the quality of the prompts into which class names are embedded (Radford et al., 2021; Pratt et al., 2022; Zhou et al., 2022b;a; Huang et al., 2022), surprisingly little attention has been paid to improving the *richness of the class names themselves*. This can be particularly crucial in cases where class names are not very informative or are too coarsely-defined to match the sort of descriptions that might arise in natural captions. Consider, for an example, the classes "large man-made outdoor things" and "reptiles" in the CIFAR20 dataset (Krizhevsky, 2009).

In this paper, we introduce a new method to tackle zero-shot classification with CLIP models for problems with coarsely-defined class labels. We refer to our method as Classification with Hierarchical Label Sets (CHiLS for short). Our method utilizes a hierarchical map to convert each class into a list of subclasses, performs normal CLIP zero-shot prediction across the union set of all *subclasses*, and finally uses the inverse mapping to convert the subclass prediction to the requi-

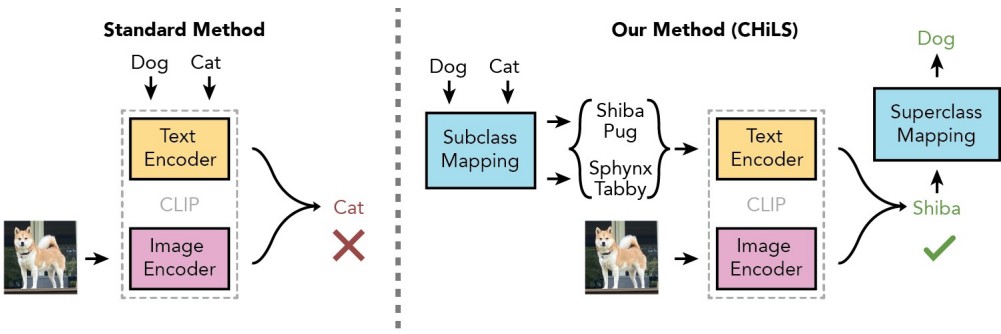

Figure 1: **(Left)** *Standard CLIP Pipeline for Zero-Shot Classification*. For inference, a standard CLIP takes in input a set of classes and an image where we want to make a prediction and makes a prediction from that set of classes. **(Right)** *Our proposed method CHiLS for leveraging hierarchical class information into the zero-shot pipeline*. We map each individual class to a set of subclasses, perform inferences in the subclass space (i.e., union set of all subclasses), and map the predicted subclass back to its original superclass.

site superclass. We additionally include a reweighting step wherein we leverage the raw superclass probabilities in order to make our method robust to less-confident predictions at the superclass and subclass level.

We evaluate CHiLS on a wide array of image classification benchmarks with *and* without available hierarchical information. In the former case, leveraging preexisting hierarchies leads to strong accuracy gains across all datasets. In the latter, we show that rather than enumerating the hierarchy by hand, using GPT-3 to query a list of *possible* subclasses for each class (whether or not they are actually present in the dataset) still leads to consistent improved accuracy over raw superclass prediction. We summarize our main contributions below:

- We propose CHiLS, a new method for improving zero-shot CLIP performance in scenarios with ill-defined and/or overly general class structures, which requires no labeled data or training time and is flexible to both existing and synthetically generated hierarchies.
- We show that CHiLS consistently performs as well or better than standard practices in situations with only synthetic hierarchies, and that CHiLS can achieve up to 30% accuracy gains when ground truth hierarchies are available.

## 2 RELATED WORK

### 2.1 TRANSFER LEARNING

While the focus of this paper is to improve CLIP models in the zero-shot regime, there is a large body of work exploring improvements to CLIP's few-shot capabilities. In the standard fine-tuning paradigm for CLIP models, practitioners discard the text encoder and only use the image embeddings as inputs for some additional training layers. This however, leads to certain problems.

One particular line of work on improving the fine-tuned capabilities of CLIP models leverages model weight interpolation. Wortsman et al. (2021) proposes to linear interpolate the weights of a fine-tuned and a zero-shot CLIP model to improve the fine-tuned model under distribution shifts. This idea is extended by Wortsman et al. (2022) into a general purpose paradigm for ensembling models' weights in order to improve robustness. Ilharco et al. (2022) then builds on both these works and puts forth a method to "patch" fine-tuned and zero-shot CLIP weights together in order to avoid the issue of catastrophic forgetting. Among all the works in this section, our paper is perhaps most similar to this vein of work (albeit in spirit), as CHiLS too seeks to combine two different predictive methods. Ding et al. (2022) also tackles catastrophic forgetting, though they propose an orthogonal direction and fine-tune both the image encoder and the text encoder, where the latter draws from a replay vocabulary of text concepts from the original CLIP database.

There is another line of work that seeks to improve CLIP models by injecting a small amount of learnable parameters into the frozen CLIP backbone. This has been commonly achieved through the adapter framework (Houlsby et al., 2019) from parameter-efficient learning; specifically, in Gao et al. (2021) they fine-tune a small number of additional weights on top of the encoder blocks, which is then connected with the original embeddings through residual connections. Zhang et al. (2021a) builds on this method by removing the need for additional training and simply uses a cached model. In contrast to these works, Jia et al. (2022) forgoes the adapter framework when using a Vision Transformer backbone for inserting learnable "prompt" vectors into the transformer's input layers, which shows superior performance over the aforementioned methods.

Additionally, some have looked at circumventing the entire process of prompt engineering. Zhou et al. (2022a) and Zhou et al. (2022b) tackle this by treating the tokens within each prompt as learnable vectors, which are then optimized within only a few images per class. Huang et al. (2022) echoes these works, but instead does not utilize any labeled data and learns the prompt representations in an unsupervised manner. Zhai et al. (2022) completely forgoes the notion of fine-tuning in the first place, instead proposing to reframe the pre-training process as only training a language model to match a pre-trained and frozen image model. In all the above situations, *some* amount of data, whether labeled or not, is used in order to improve the predictive accuracy of the CLIP model.

## 2.2 ZERO-SHOT PREDICTION

The field of Zero-Shot Learning (ZSL) has existed well before the emergence of open vocabularly models, with its inception traced to Larochelle et al. (2008). With regards to non-CLIP related methods, the ZSL paradigm has shown success in improving multilingual question answering (Kuo & Chen, 2022) with large language models (LLMs), and also in image classification tasks where wikipedia-like context is used in order to perform the classification without access to the training labels (Bujwid & Sullivan, 2021; Shen et al., 2022).

With CLIP models, ZSL success has been found in a variety of tasks. Namely, Zhang et al. (2021b) expands the CLIP 2D paradigm for 3D point clouds. Tewel et al. (2021) shows that CLIP models can be retrofitted to perform the reverse task of image-to-text generation, and Shen et al. (2021) likewise display's CLIP's ability to improve performance on an array of Vision&Language tasks. Both Yu et al. (2022) and Cho et al. (2022) expand CLIP's zero-shot abilities through techniques drawn from reinforcement learning (RL), with the former using CLIP for the task of audio captioning. Gadre et al. (2022) similarly works with the RL literature and retrofits CLIP to improve the embodied AI task of object navigation without any additional training. Zeng et al. (2022) shows the capabilities of composing CLIP-like models and LLMs together to extend the zero-shot capabilities to tasks like assitive dialogue and open-ended reasoning. Unlike our work here, these prior directions mostly focus on generative problems or, in the case of Bujwid & Sullivan (2021) and Shen et al. (2022), require rich external knowledge databases to employ their methods.

In the realm of improving CLIP's zero-shot capabilities for image classification, we particularly note the contemporary work of Pratt et al. (2022). Here, authors explore using GPT-3 to generate rich textual prompts for each class rather than using preexisting prompt templates, and show improvements in zero-shot accuracy across a variety of image classification baselines. In another work, Ren et al. (2022) proposes leveraging preexisting captions in order to improve performance, though this is restricted to querying the pre-training set of captions. In contrast, our work explores a complementary direction of leveraging hierarchy in class names to improve zero-shot performance of CLIP. with a fixed set of preexisting prompt templates.

## 2.3 HIERARCHICAL CLASSIFICATION

Our present work is related to the domain of Hierarchical Classification (Silla & Freitas, 2010), i.e. classification tasks when the set of labels can be arranged in a DAG-like class hierarchy. Methodologies from this domain have been extensively used for multi-label classification (Dimitrovski et al., 2011; Liu et al., 2021; Chalkidis et al., 2020), and recent works have shown that this paradigm can aid in zero-shot learning by attempting to uncover hierarchical relations between classes (Chen et al., 2021; Mensink et al., 2014) and/or leveraging existing hierarchical information during training (Yi et al., 2022; Cao et al., 2020). Our line of inquiry is orthogonal to these approaches, as CHiLS is

---

**Algorithm 1** **C**lassification with **Hi**erarchical **L**abel **S**ets (CHiLS)

---

**input** : data point $x$, class labels $\mathcal{C}$, prompt function T, label set mapping $G$, CLIP model $f$

1: Set $\mathcal{C}_{\text{sub}} \leftarrow \cup_{c_i \in \mathcal{C}} G(c_i)$         ▷ Union of subclasses for subclass prediction

2: $\widehat{y}_{\text{sub}} = \sigma(f(x, \text{T}(\mathcal{C}_{\text{sub}})))$         ▷ Subclass probabilities

3: $\widehat{y}_{\text{sup}} = \sigma(f(x, \text{T}(\mathcal{C})))$         ▷ Superclass probabilities

4: **for** $i = 1$ to $|\mathcal{C}|$ **do**

5:     $S_{c_i} = G(c_i)$

6:     **for** $s_{c_i,j} \in S_{c_i}$ **do**

7:        $\widehat{y}_{\text{sub}}[s_{c_i,j}] = \widehat{y}_{\text{sub}}[s_{c_i,j}] * \widehat{y}_{\text{sup}}[c_i]$

               ▷ Combining subclass and superclass prediction probability

8:     **end for**

9: **end for**

**output** : $G^{-1}(\arg\max \widehat{y}_{\text{sub}})$

---

easily composable into standard CLIP pipelines, only models unseen class hyponyms (rather than relationships between known classes), and requires no training of hierarchical embeddings.

## 3   Proposed Method

In this paper, we are primarily concerned with the problem of zero-shot image classification in CLIP models (see App. B for an introduction to CLIP and relavent terminology). For CLIP models, zero-shot classification involves using both a pretrained image encoder and a pretrained text encoder (see the left part of Figure 1). To perform a zero-shot classification, we need a predefined set of classes written in natural language. Let $\mathcal{C} = \{c_1, c_2, \ldots, c_k\}$ be such a set. Given an image and set of classes, each class is embedded within a natural language prompt (through some function $\text{T}(\cdot)$) to produce a "caption" for each class (e.g. one standard prompt mentioned in Radford et al. (2021) is *"A photo of a {}."*). These prompts are then fed into the text encoder and after passing the image through the image encoder, we calculate the cosine similarity between the image embedding and each class-prompt embedding. These similarity scores form the output "logits" of the CLIP model, which can be passed through a softmax to generate the class probabilities.

As noted in Section 2, previous work has focused on improving the $\text{T}(\cdot)$ for each class label $c_i$. With CHiLS, we instead focus on the complementary task of directly modifying the set of classes $\mathcal{C}$ when $\mathcal{C}$ is ill-formed or overly general, while keeping $\text{T}(\cdot)$ fixed. Our method involves into two main steps: (1) using hierarchical information to perform inference across *subclasses*, and (2) leveraging raw superclass probabilities to combine the best of subclass and superclass prediction probabilities.

### 3.1   Zero-Shot Prediction with Hierarchical Label Sets

Our method CHiLS slightly modifies the standard approach for zero-shot CLIP prediction. As each class label $c_i$ represents some concept in natural language (e.g. the label "dog"), we acquire a **subclass set** $\mathcal{S}_{c_i} = \{s_{c_i,1}, s_{c_i,2}, \ldots, s_{c_i,m_i}\}$ through some mapping function $G$, where each $s_{c_i,j}$ is a linguistic *hyponym*, or subclass, of $c_i$ (e.g. corgi for dogs) and $m_i$ is the size of the set $S_{c_i}$.

Given a label set $S_{c_i}$ for each class, we proceed with the standard process for zero-shot prediction, but now using the *union* of all label sets as the set of classes. Through this, CHiLS will now produce its guess for the most likely *subclass*. We then leverage the inverse mapping function $G^{-1}$ to coarse-grain our prediction back into the corresponding superclass. Our method is detailed more formally in Algorithm 1.

In our work, we experiment with two scenarios: (i) when hierarchy information is available and can be readily queried; and (ii) when hierarchy information is *not* available and the label set for each class must be generated, which we do so by prompting GPT-3.

### 3.2   Reweighting probabilities with Superclass Confidence

While the above method is able to effectively utilize CLIP's ability to identify relatively fine-grained concepts, by predicting on only subclass labels we lose any positive benefits of the superclass label,

| Dataset | Superclass Accuracy | CHiLS Accuracy (Existing Map) | CHiLS Accuracy (GPT-3 Map) |
|---|---|---|---|
| Nonliving26 | 79.82 | 90.67 (+10.85) | 81.51 (+1.69) |
| Living17 | 91.08 | 93.80 (+2.72) | 91.43 (+0.35) |
| Entity13 | 77.46 | 92.59 (+15.13) | 78.11 (+0.65) |
| Entity30 | 70.32 | 88.87 (+18.55) | 71.75 (+1.43) |
| CIFAR20 | 59.54 | 85.30 (+25.76) | 65.90 (+6.36) |
| Food-101 | 91.82 | N/A | 91.73 (−0.09) |
| Fruits-360 | 60.47 | 60.87 (+0.40) | 62.17 (+1.70) |
| Fashion1M | 45.78 | N/A | 47.44 (+1.66) |
| Fashion-MNIST | 68.50 | N/A | 70.85 (+2.35) |
| LSUN-Scene | 88.20 | N/A | 88.97 (+0.77) |
| Office31 | 89.13 | N/A | 89.37 (+0.24) |
| OfficeHome | 88.85 | N/A | 88.76 (−0.09) |
| ObjectNet | 53.10 | 85.34 (+32.24) | 53.52 (+0.42) |
| EuroSAT | 62.10 | N/A | 62.40 (+0.30) |
| RESISC45 | 72.13 | N/A | 72.52 (+0.40) |

Table 1: Zero-shot accuracy performance across image benchmarks with superclass labels (baseline), CHiLS with existing hierarchy (whenever available), and CHiLS with GPT-3 generated hierarchy. CHiLS improves classification accuracy in all situations with given label sets and all but 2 datasets with GPT-3 generated label sets.

and performance may vary widely based on the quality of the subclass labels. Given recent evidence (Minderer et al., 2021; Kadavath et al., 2022) that large language models (like the text encoder in CLIP) are well-calibrated and generally predict correct labels with *high* probability, we modify our initial algorithm to leverage this behavior and utilize *both* superclass and subclass information. We provide empirical evidence of this property in Appendix A.

Specifically, we include an additional reweighting step within our main algorithm (see lines 4-9 in Algorithm 1). Here, we reweight each set of subclass probabilities by its superclass probability. Heuristically, as the prediction is now taken as the argmax over *products* of probabilities, large disagreements between subclass and superclass probabilities will be down-weighted (especially if one particular superclass is confident) and subclass probabilities will be more important in cases where the superclass probabilities are roughly uniform. We show ablations on the choice of the reweighting algorithm in Section 4.4.

## 4 EXPERIMENTS

In this section, we first lay out the experimental set-up. We then discuss, in order, the efficacy of our proposed method in situations with available class hierarchy information and in situations *without* any preexisting hierarchy. After these main results, we present a series of ablations over various design choices showing where our method is robust and what might be crucial for its performance.

### 4.1 SETUP

**Datasets.** As we are primarily concerned with improving zero-shot CLIP performance in situations with ill-formed and/or semantically coarse class labels, we test our method on the following image benchmarks: the four BREEDS imagenet subsets (living17, nonliving26, entity13, and entity30) (Santurkar et al., 2021), CIFAR20 (the coarse-label version of CIFAR100) (Krizhevsky, 2009), Food-101 (Bossard et al., 2014), Fruits-360 (Mureşan & Oltean, 2018), Fashion1M (Xiao et al., 2015), Fashion-MNIST (Xiao et al., 2017), LSUN-Scene (Yu et al., 2015), Office31 (Saenko et al., 2010), OfficeHome (Venkateswara et al., 2017), ObjectNet (Barbu et al., 2019), EuroSAT (Helber et al., 2019; 2018), and RESISC45 (Cheng et al., 2017). These datasets constitute a wide range of different image domains and include datasets with and without available hierarchy information. Additionally, the chosen datasets vary widely in the semantic granularity of their classes, from overly general cases (CIFAR20) to settings with a mixture of general and specific classes (Food-101, OfficeHome). We also examine CHiLS's robustness to distribution shift within a dataset by averaging all results for the BREEDS datasets, Office31, and OfficeHome across different shifts (see

Appendix H for more information). We additionally modify the Fruits-360 and ObjectNet datasets to create existing taxonomies. More details for dataset preparation are detailed in Appendix H.

**Model Architecture.** Unless otherwise specified, we use the ViTL/14@336px backbone (Radford et al., 2021) for our CLIP model, and used DaVinci-002 (with temperature fixed at 0.7) for all ablations involving GPT-3. For the choice of the prompt embedding function $T(\cdot)$, for each dataset we experiment (where applicable) with two different functions: (1) Using the average text embeddings of the 75 different prompts for each label used for ImageNet in Radford et al. (2021), where the prompts cover a wide array of captions and (2) Following the procedure that Radford et al. (2021) puts forth for more specialized datasets, we modify the standard prompt to be of the form *"A photo of a {}, a type of [context]."*, where **[context]** is dataset-dependent (e.g. "food" in the case of food-101). In the case that a custom prompt set exists for a dataset, as is the case with multiple datasets that the present work shares with Radford et al. (2021), we use the given prompt set for the latter option rather than building it from scratch. For each dataset, we use the prompt set that gives us the best *baseline* (i.e. superclass) zero-shot performance. More details are in Appendix C.

**Choice of Mapping Function $G$.** In our experiments, we primarily look at how the choice of the mapping function $G$ influences the performance of CHiLS. In Section 4.2, we focus on the datasets with available hierarchy information. Here, $G$ and $G^{-1}$ are simply table lookups to find the list of subclasses and corresponding superclass respectively. In Section 4.3, we explore situations in which the true set of subclasses in each superclass is unknown. In these scenarios, we use GPT-3 to generate our mapping function $G$. Specifically, given some label set size $m$, superclass name `class-name`, and optional **context** (which we use whenever using the context-based prompt embedding), we query GPT-3 with the prompt:

```
Generate a list of m types of the following [context]:  class-name
```

The resulting output list from GPT-3 thus defines our mapping $G$ from superclass to subclass. Unless otherwise specified, we fix $m = 10$ for all datasets. Additionally, in 4.4 we explore situations in which hierarchical information is present but noisy, i.e. the label set for each superclass contains the true subclasses *and* erroneous subclasses that are not present in the dataset.

## 4.2 Leveraging Available Hierarchy Information.

We first concern ourselves with the scenario in which there is hierarchy information already available (or readily accessible) for a given dataset. In this situation, the set of subclasses for each superclass is exactly specified and correct (i.e. every image within each superclass falls into one of the subclasses). We emphasize that here we do *not* need information about which example belongs to which subclass, we just need a mapping of superclass to subclass. For example, each class in the BREEDS dataset living17 is made up of 4–8 ImageNet subclasses at finer granularity (e.g. "parrot" includes "african grey" and "macaw").

**Results** In Table 1, we can see that our method performs better than using the baseline superclass labels alone across all 7 of the datasets with available hierarchy information, in some cases leading to +15% improvement in predictive accuracy.

## 4.3 CHiLS in Unknown Hierarchy Settings

Though we have seen considerable success in situations with access to the true hierarchical structure, in some real-world settings our dataset may not include any available information about the subclasses within each class. In this scenario, we turn to using GPT-3 to approximate the hierarchical map $G$ (as specified in Section 4.1). It is important to note that GPT-3 may sometimes output suboptimal label sets, most notably in situations where GPT-3 chooses the wrong wordsense or when GPT-3 only lists modifiers on the original superclass (e.g. producing the list [red, yellow, green] for types of apples). In order to account for these issues in an out-of-the-box fashion, we automatically append the superclass name (if not already present) to each generated subclass label, and also include the superclass itself within the label set. For a controlled analysis about the effect of including the superclass itself in the label set, see Appendix D.

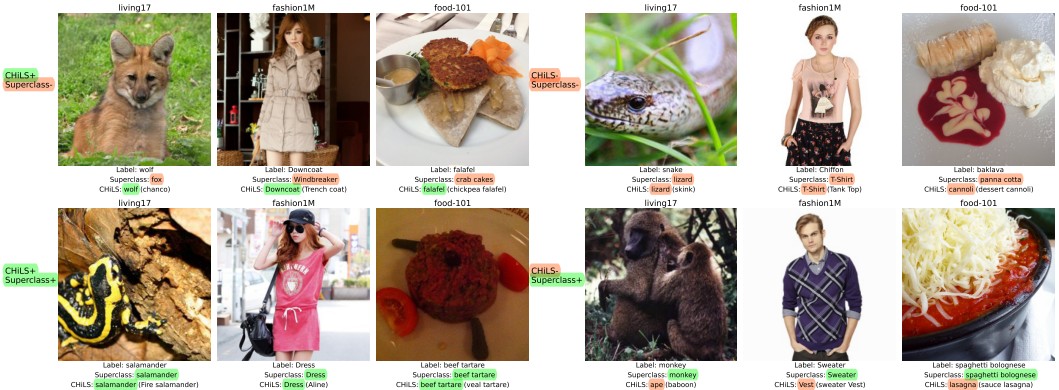

Figure 2: Selected examples of behavior differences between the superclass performance (which is our baseline) and CHiLS performance across three different datasets. (Upper left): CHiLS is correct, superclass prediction is not. (Lower left): Both correct. (Upper right): Both wrong. (Lower Right): superclass prediction is correct, CHiLS is not.

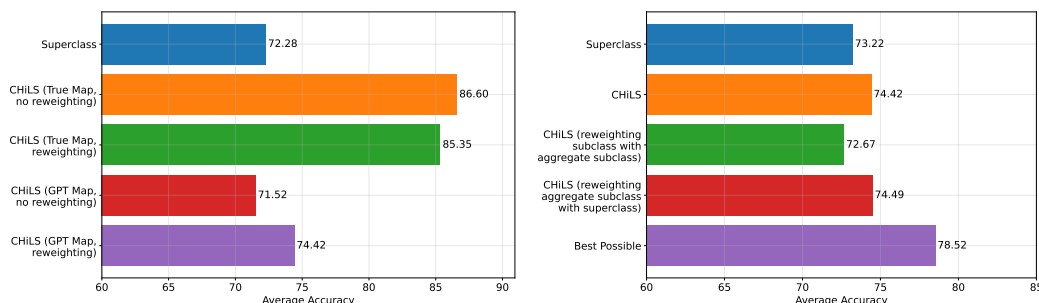

Figure 3: (Left) Average accuracy across datasets for superclass prediction, CHiLS (ours), and CHiLS *without* the reweighting step. While when given the true hierarchy omitting the reweighting step can slightly boost performance beyond CHiLS, in situations without the true hierarchy the reweighting step is crucial to improving on the baseline accuracy. (Right) Average accuracy across datasets with GPT-generated label sets for different reweighting algorithms. Using aggregate subclass probabilities for reweighting performs noticeably worse than our initial method and reweighting in superclass space. CHiLS too only performs slightly worse than the contrived best possible union of subclass and superclass predictions.

**Results** In this setting, our method is still able to beat the baseline performance in most datasets, albeit with lower accuracy gains (see Table 1). Thus, while knowing the true subclass hierarchy can lead to large accuracy gains, it is enough to simply enumerate a list of possible subclasses for each class with no prior information about the dataset in order to improve the predictive accuracy. In Figure 2, we show selected examples to highlight CHiLS's behavior across three datasets.

## 4.4 ABLATIONS

**Is Reweighting Necessary?** Though the reweighting step in CHiLS is motivated by the evidence that CLIP generally assigns higher probability to *correct* predictions rather than incorrect ones (see Appendix A for empirical verification), it is not immediately clear whether it is truely necessary. Averaged across all documented datasets, in Figure 3 (left) we show that in the true hierarchy setting, not reweighting the subclass probabilities can actually slightly *boost* performance (as the label sets are adequately tuned to the distribution of images). However, in situations where the true hierarchy is not present, omitting the reweighting step puts accuracy below the baseline performance. We attribute this difference in behavior to the fact that reweighting multiplicatively combines the superclass and subclass predictions, and thus if subclass performance is sufficient on its own (as is the case when the true hierarchy is available) then combining it with superclass predictions can

| Dataset | Superclass Accuracy | CHiLS Accuracy (Existing Map) | CHiLS Accuracy (Existing Map + Noise) |
|---------|---------------------|-------------------------------|---------------------------------------|
| nonliving26 | 79.82 | 90.67 (+10.85) | 89.48 (+9.66) |
| living17 | 91.08 | 93.80 (+2.72) | 92.47 (+1.39) |
| entity13 | 77.46 | 92.59 (+15.13) | 90.34 (+12.88) |
| entity30 | 70.32 | 88.87 (+18.55) | 87.56 (+17.24) |

Table 2: CHiLS zero-shot accuracy when $G$ includes *all* subclasses in the ImageNet hierarchy descended from the respective root node. Even in the presence of noise added to the true label sets, CHiLS is able to make large accuracy gains.

cause the model to more closely follow the behavior of the underperforming superclass predictor. Thus, as not having the true hierarchy is considerably more likely in the wild, the reweighting step is imperative to utilizing CHiLS to its fullest.

**Different Reweighting Strategies**   We too experimented on whether the initial reweighting algorithm is the optimal method for combining superclass and subclass predictions. Namely, we investigated whether superclass probabilities could be replaced by the sum over the matching subclass probabilities, *and* whether we can aggregate subclass probabilities and reweight them with the matching superclass probabilities (i.e. performing the normal reweighting step but in the space of superclasses). In Figure 3 (right) we show that replacing the superclass probabilities in the reweighting step with aggregate subclass probabilities removes any accuracy gains from CHiLS, but that doing the reweighting step in superclass space *does* maintain CHiLS accuracy performance. This suggests that the beneficial behavior of CHiLS may be due to successfully combining two different sets of class labels. We also display the upper bound for combining superclass and subclass prediction (i.e. the accuracy when a datum is correctly labeled if the superclass *or* subclass predictions are correct) in purple, which we note is impossible in practice, and observe that even the best possible performance is not much higher than the performance of CHiLS.

**Noisy Available Hierarchies**   While the situation described in Section 4.3 is the most probable in practice, we additionally investigate the situation in which the hierarchical information is present but *overestimates* the set of subclasses. For example, the scenario in which a dataset with the class "dog" includes huskies and corgis, but CHiLS is provided with huskies, corgis, *and Labradors* as possible subclasses, with the last being out-of-distribution. To do this, we return to the BREEDS datasets presented in Santurkar et al. (2021). As the BREEDS datasets were created so that each class contains the same number of subclasses (which are ImageNet classes), we modify $G$ such that the label set for each superclass corresponds to *all* the ImageNet classes descended from that node in the hierarchy (see Appendix G for more information). As we can see in Table 2, CHiLS is able to improve upon the baseline performance even in the presence of added noise in each label set.

**Label Set Size**   In previous works investigating importance of prompts in CLIP's performance, it has been documented that the number of prompts used can have a decent effect on the overall performance (Pratt et al., 2022; Santurkar et al., 2022). Along this line, we investigate how the size of the *subclass set* generated for each class effects the overall accuracy by re-running our main experiments with varying values of $m$ (namely, 1, 5, 10, 15, and 50). In Figure 4 (right), there is little variation across label set sizes that is consistent over all datasets, though $m = 1$ has a few very low performing outliers due to the extremely small label set size. We observe that the optimal label set size is context-specific, and depends upon the total number of classes present and the semantic granularity of the classes themselves. Individual dataset results are available in Appendix E.

**Model Size**   In order to examine whether the performance of CHiLS only exists within the best performing CLIP backbone (e.g. ViT-L/14@336), we measure the average relative change in accuracy performance between CHiLS and the baseline superclass predictions across all datasets for an array of different CLIP models. Namely, we investigate the RN50, RN101, RN50x4, ViT-B/16, ViT-B/32, and ViT-L/14@336 CLIP backbones (see Radford et al. (2021) for more information on the model specifications). In Figure 4 (left), we show that across the 6 specified CLIP backbones, CHiLS performance leads to relatively consistent relative accuracy gains, with a slight (but not confidently significant) trend showing improved performance for the ResNet backbones over the ViT

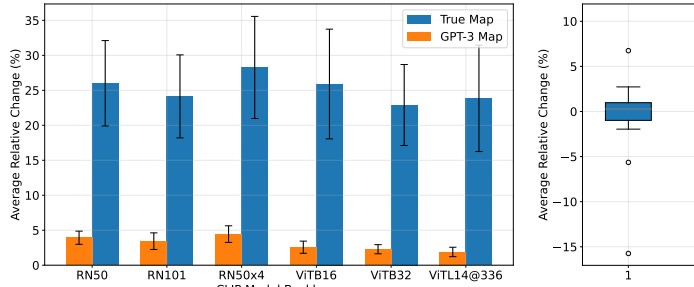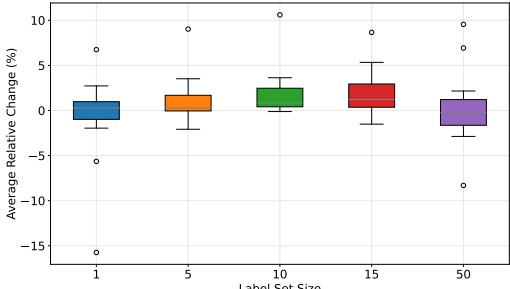

Figure 4: (Left) Average relative change between CHiLS and baseline for true mapping and GPT-3 generated mapping. Across changes in CLIP backbone size and structure, the effectiveness of CHiLS at improving performance only varies slightly. (Right) Average relative accuracy change from the baseline to CHiLS (across all datasets), for varying label set sizes. In all, there is not much difference in performance across label set sizes.

backbones, which is to be expected given their worse base capabilities. This shows that CHiLS's benefits are not an artifact of large model size.

**Alternative Aggregating Methods** While CHiLS is based on a *set-based* mapping approach for subclasses and a linear averaging for prompt templates (based on Radford et al. (2021)'s procedure), we experimented with two alternative ensembling methods for different parts of the CHiLS pipeline: (1) Using a *linear average* of subclass embeddings rather than the set-based mapping (that is, every superclass's text embedding is the average across all subclass embeddings, each themselves averaged across every prompt template) and (2) Using a *set-based* mapping for prompt templates rather than a linear average (i.e. instead of averaging across prompt templates, predict across each prompt template separately at inference time and then use embedded class to map back to the set of superclasses). Note in the latter case we only experiment with how this effects *superclass* prediction (where each class maps to a set of the dataset's chosen prompt embeddings), as using set-based ensembling for *both* prompts and subclasses within CHiLS quickly becomes computationally expensive. In Figure 6 (in Appendix F), we see that using our initial aggregation methods (i.e. linear averaging for prompts and set mappings for subclasses) achieves greater accuracy.

## 5 CONCLUSION

In this work, we demonstrated that the zero-shot image classification capabilities of CLIP models can be improved by leveraging hierarchical information for a given set of classes. When hierarchical structure is available in a given dataset, our method shows large improvements in zero-shot accuracy, and even when subclass information *isn't* explicitly present, we showed that we can leverage GPT-3 to generate subclasses for each class and still improve upon the baseline (superclass) accuracy.

We remark that CHiLS may be quite beneficial to practitioners using CLIP as an out-of-the-box image classifier. Namely, we show that in scenarios where the class labels may be ill-formed or overly coarse, even without existing hierarchical data accuracy can be improved with a *fully automated* pipeline (via querying GPT-3), yet CHiLS is flexible enough that any degree of hand-crafting label sets can be worked into the zero-shot pipeline. Our method has the added benefit of being both *completely zero-shot* (i.e. no training or fine-tuning necessary) and is resource efficient.

**Limitations and Future Work** As with usual zero-shot learning, we don't have a way to validate the performance of our method. Additionally, we recognize that CHiLS is suited for scenarios in which a semantic hierarchy most likely exists, and thus may not be particularly useful in classification tasks where the classes are already decently fine-grained. We believe that this limitation will not hinder the applicability of our method, as practitioners would know if their task contains any latent semantic hierarchy and thus choose to use our method or not a priori. Given CHiLS's empirical successes, we hope to perform more investigation to develop an understanding of *why* CHiLS is able to improve zero-shot accuracy and whether there is a more principled way of reconciling superclass and subclass predictions.

## REPRODUCIBLITY STATEMENT

The source code for reproducing the work presented here is available at `https://github.com/anonOpenReview1/clip-hierarchy`. We implement our method in PyTorch (Paszke et al., 2017) and provide an infrastructure to run all the experiments to generate corresponding results. We have stored all models and logged all hyperparameters and seeds to facilitate reproducibility. Additionally, all necessary data preprocessing details are present in Appendix H.

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

APPENDIX

# A EMPIRICAL EVIDENCE OF CLIP CONFIDENCE

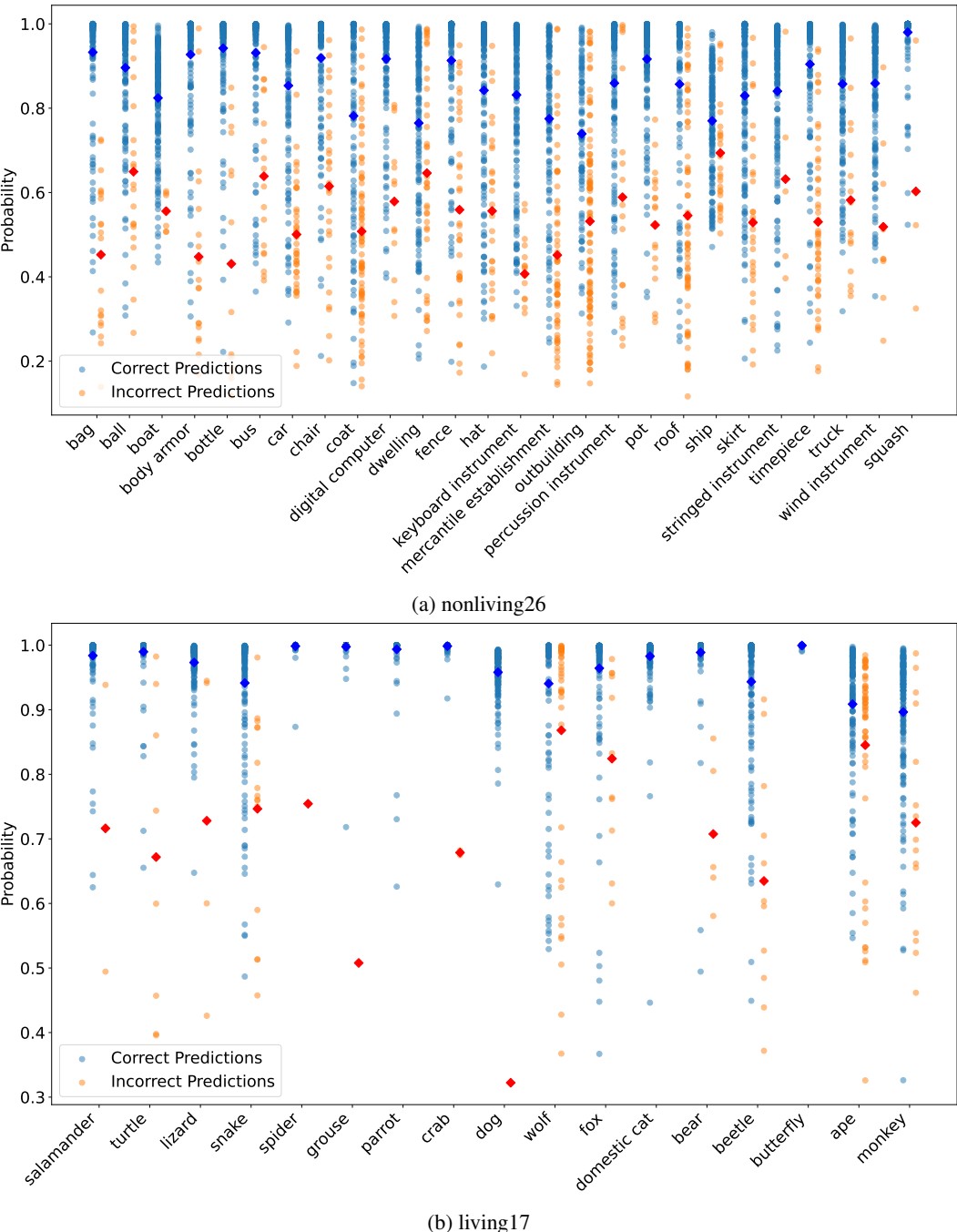

(a) nonliving26

(b) living17

Figure 5: Distribution of argmax probabilities across ImageNet BREEDS datasets for correctly and incorrectly classified data points, with the diamonds representing average probability for each class. Correctly classified probabilities are on average higher than the misclassified probabilities.

The motivation behind the reweighting step of CHiLS primarily comes from the heuristic that LLMs make correct predictions with high estimated probabilities assigned to them (Kadavath et al., 2022), and that CLIP models themselves are well-calibrated (Minderer et al., 2021). However, we also

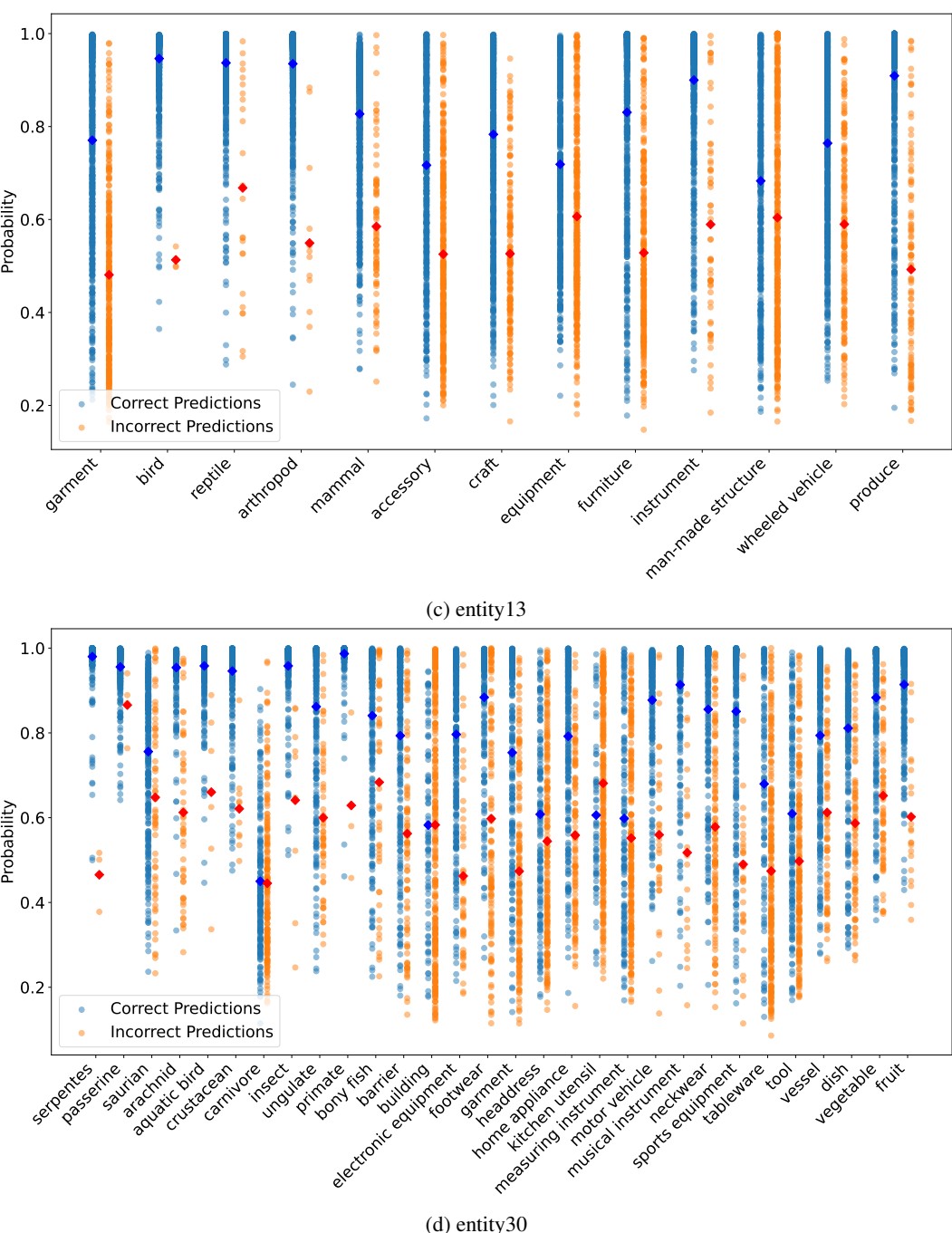

(c) entity13

(d) entity30

verify whether there is some evidence of this behavior in CLIP models. Given that the output of a CLIP model is a probability distribution over the provided classes, we care specifically about the probability of the *argmax* class (i.e. the predicted class) when the model is correct and when it is incorrect. Across the BREEDS datasets for the standard ImageNet domain, in Figure 5 we show the distribution of the correct and incorrect argmax probabilities for each class (i.e. for each class $c_i$, we show the output probabilties for $c_i$ when it was correctly classified and the output probabilities of the predicted classes when the true class is $c_i$). Whenever CLIP is correct, the associated probability is on average much higher than the probabilities associated with misclassification.

## B CLIP PRIMER

*Open Vocabulary models* (as termed in Pham et al. (2021)) refer to models that are able to classify images by associating them with natural language descriptions of each class. These models are "open" in the sense that they are to predict on an *arbitrary* vocabulary of descriptions (as opposed to a fix set), thus allowing for arbitrary-way image classification. Popular open vocabulary models include the model of focus CLIP (Radford et al., 2021) and ALIGN (Jia et al., 2021) as examples.

**C**ontrastive **L**anguage **I**mage **P**retraining (CLIP) is a family of open vocabulary models, and the focus of the present work. CLIP, which is comprised of a text encoder and an image encoder that project into the same latent space, is trained in the following way: Given a set of image-caption pairs (e.g. a photo of a dog with the caption "a photo of a dog."), CLIP is trained to predict which caption goes with which image as a contrastive learning objective by comparing the similarity between each image embedding and each caption embedding.

At inference time (in the zero-shot setting), a naïve method for image classification (which is the initial baseline tried in Radford et al. (2021)) involves simply passing in the list of class names for a given dataset, and calculating the similarity between a particular image embedding and each one of these class embeddings. However, Radford et al. (2021) found that by taking a cue from the recent literature on prompt engineering for large language models (Gao et al., 2020), CLIP can perform significantly better as a zero-shot predictor if each class name is included in a natural language **prompt** that resembles some sort of image caption (as that is what CLIP was trained on). As an example, the standard baseline prompt mentioned is *"A photo of a {}."*. In our work, we define a prompt (or prompt template, which we use interchangeably) as any caption-like phrase in natural language that a class name can be injected into.

## C ADDING CONTEXT TO PROMPTS AND GPT-3 QUERIES

| Dataset | [context] | Prompt Set Used |
|---------|-----------|-----------------|
| Nonliving26 | N/A | ImageNet |
| Living17 | N/A | ImageNet |
| Entity13 | N/A | ImageNet |
| Entity30 | N/A | ImageNet |
| CIFAR20 | N/A | ImageNet |
| Food-101 | "food" | Dataset-Specific |
| Fruits-360 | "fruit" | Dataset-Specific |
| Fashion1M | "article of clothing" | Dataset-Specific |
| Fashion-MNIST | "article of clothing" | ImageNet |
| LSUN-Scene | N/A | ImageNet |
| Office31 | "office supply" | Dataset-Specific |
| OfficeHome | "office supply" | ImageNet |
| ObjectNet | N/A | ImageNet |
| EuroSAT | N/A | Dataset-Specific |
| RESISC45 | N/A | Dataset-Specific |

Table 3: Context tokens and prompt sets used for each dataset.

In order to disentangle the effect that well-formed prompt templates have on the success of CHiLS, for each dataset (besides the BREEDS datasets and ObjectNet as they are already semantically similar to ImageNet) we compare the ImageNet 75 classes against a dataset-specific set of prompt templates. In the case of EuroSAT, RESISC45, CIFAR20 and Food-101, we directly use the prompt template set from Radford et al. (2021). For LSUN-Scene, we use the prompt template set for SUN397 (Xiao et al., 2010), as the two datasets are semantically similar. For the rest of the datasets not yet mentioned (namely Fruits360, Fashion1M, Fashion-MNIST, Office31, and OfficeHome) we add the **[context]** marker into the standard prompt template as mentioned in Section 4.1. The prompt sets themselves can be directly found in the code implementation for this project.

For the GPT-3 Query with additional context, we add the respective **[context]** token to the query *if* the dataset-specific prompt template is used. Note that we did not create **[context]** tokens for EuroSAT, LSUN-Scene, or RESISC45 despite testing dataset-specific prompt templates, as there did not seem to be a concise semantic label to describe the classes in these datasets. In Table 3, we list the dataset, the **[context]** token (if applicable), and the final prompt set used for all the experiments. Here, we found that while dataset-specific prompts often improved baseline performance, they were not *gauranteed* to improve performance, as in both Fasion-MNIST and OfficeHome the general ImageNet prompt set performed better.

## D  INCLUDING SUPERCLASS LABELS IN LABEL SETS

| Dataset | CHiLS Accuracy (Existing Map) | CHiLS Accuracy (Existing Map+) | CHiLS Accuracy (GPT-3 Map) | CHiLS Accuracy (GPT-3 Map+) |
|---|---|---|---|---|
| Nonliving26 | 90.67 (+10.85) | 89.80 (+9.98) | 81.46 (+1.63) | 81.51 (+1.69) |
| Living17 | 93.80 (+2.72) | 93.62 (+2.54) | 91.30 (+0.22) | 91.43 (+0.35) |
| Entity13 | 92.59 (+15.13) | 92.06 (+14.60) | 76.97 (−0.48) | 78.11 (+0.65) |
| Entity30 | 88.87 (+18.55) | 87.29 (+16.97) | 71.80 (+1.48) | 71.75 (+1.43) |
| CIFAR20 | 85.30 (+25.76) | 81.40 (+21.86) | 65.70 (+6.16) | 65.90 (+6.36) |
| Food-101 | N/A | N/A | 91.63 (−0.19) | 91.73 (−0.09) |
| Fruits-360 | 60.87 (+0.40) | 60.63 (+0.16) | 62.48 (+2.01) | 62.17 (+1.70) |
| Fashion1M | N/A | N/A | 47.51 (+1.73) | 47.44 (+1.66) |
| Fashion-MNIST | N/A | N/A | 67.82 (−0.98) | 70.85 (+2.35) |
| LSUN-Scene | N/A | N/A | 88.80 (+0.60) | 88.97 (+0.77) |
| Office31 | N/A | N/A | 86.58 (−2.71) | 89.37 (+0.24) |
| OfficeHome | N/A | N/A | 87.88 (−0.97) | 88.76 (−0.09) |
| ObjectNet | 85.34 (+32.24) | 81.30 (+28.20) | 51.23 (−2.07) | 53.52 (+0.42) |
| EuroSAT | N/A | N/A | 62.21 (+0.11) | 62.40 (+0.30) |
| RESISC45 | N/A | N/A | 71.64 (-0.48) | 72.52 (+0.40) |

Table 4: Zero-Shot Accuracy Performance across benchmarks, controlling for the presence of the superclass label within each respective label set. In the existing map case, adding the superclass labels removes some of the performance gains of the raw existing map. In the GPT-3 Map case, adding the superclass is crucial to maintaining performance in most datasets

With CHiLS when the existing map is not available, we append the superclass name to each label set to account for possible noise in the GPT-generated label set. In Table 4, we show the effect that this inclusion has in both the existing map and GPT-map cases. Note that in the main paper, columns 1 and 4 correspond to the main results (i.e. no superclass labels in existing maps and superclass labels in GPT-3 maps). In both cases, the presence of the superclass label more effectively strikes a balance between subclass and superclass predictions. In the existing map case, this actually *hurts* performance, as the subclass labels are optimal in the given dataset. In the GPT-3 map case, while there are some datasets where removing the superclass label improves performance (namely Fruits360 and Entity30), in ever other case removing the superclass label hurts performance, sometimes by multiple percentage points.

# E    LABEL SET ABLATION ACCURACY

| Dataset | CHiLS ($m = 1$) | CHiLS ($m = 5$) | CHiLS ($m = 10$) | CHiLS ($m = 15$) | CHiLS ($m = 50$) |
|---|---|---|---|---|---|
| Nonliving26 | 80.05 (+0.23) | 81.12(+1.30) | 81.51 (+1.69) | **81.79** (+1.97) | 79.60 (−0.20) |
| Living17 | 91.47 (+0.39) | **92.69** (+1.61) | 91.43 (+0.35) | 91.55 (+0.48) | 91.52 (+0.45) |
| Entity13 | 76.75 (−0.70) | 78.15 (+0.70) | 78.11 (+0.65) | **78.42** (+0.96) | 75.82 (−1.63) |
| Entity30 | 72.25 (+1.93) | 71.47 (+1.15) | 71.75 (+1.43) | **73.38** (+3.06) | 70.18 (−0.14) |
| CIFAR20 | 63.61 (+4.02) | 64.95 (+5.41) | **65.90** (+6.36) | 62.80 (+3.26) | 63.72 (+4.13) |
| Food-101 | 91.57 (−0.25) | **91.75** (−0.07) | 91.73 (−0.09) | 91.51 (−0.31) | 91.58 (−0.24) |
| Fruits-360 | 61.18 (+0.47) | 62.33 (+1.86) | 62.17 (+1.70) | **62.51** (+2.04) | 61.19 (+0.48) |
| Fashion1M | 38.57 (−7.21) | 45.77 (−0.01) | **47.55** (+1.66) | 46.93 (+1.15) | 41.98 (−3.80) |
| Fashion-MNIST | 67.16 (−1.34) | **70.93** (+2.44) | 70.84 (+2.35) | 69.09 (+0.60) | 69.98 (+1.49) |
| LSUN-Scene | 87.20 (−1.00) | 86.33 (−1.87) | **88.97** (+0.77) | 86.80 (−1.40) | 85.60 (−2.60) |
| Office31 | 89.46 (+0.36) | 88.08 (−1.05) | 89.37 (+0.24) | 89.03 (−0.10) | **90.23** (+1.10) |
| OfficeHome | 88.06 (−0.79) | **89.12** (+0.27) | 88.76 (−0.09) | 89.06 (+0.21) | 88.39 (−0.46) |
| ObjectNet | 50.12 (−2.98) | 53.29 (+0.18) | 58.19 (+0.42) | 57.66 (+4.56) | **58.19** (+5.09) |
| EuroSAT | 62.59 (+0.49) | 62.21 (+0.10) | 62.40 (+0.30) | **62.89** (+0.79) | 61.39 (−0.71) |
| RESISC45 | **73.19** (+1.06) | 72.05 (−0.08) | 72.52 (+0.40) | 72.50 (+0.38) | 70.61 (−1.52) |

Table 5: Accuracy across different label set sizes generated by GPT-3, with best performing label set size in each row bolded. In general, there is no consistent trend related to label set size and zero-shot performance across datasets.

Table 5 displays the raw accuracy scores for CHiLS across different label set sizes.

# F    ALTERNATIVE AGGREGATION RESULTS

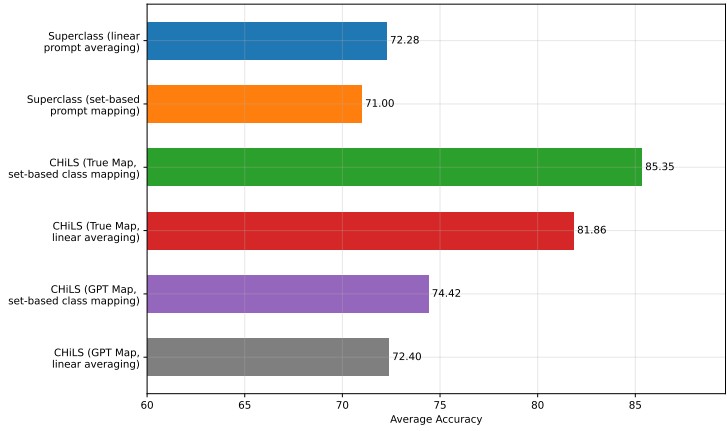

Figure 6: Average accuracy across datasets for varying aggregative methods on both the prompt and subclass steps of the zero-shot pipeline. In general, linear averaging for subclasses performs worse than our proposed set-based method, while linear averaging for prompts (for raw superclass prediction) performs better thant using a set-based mapping.

Figure 6 displays the results of our ablation on alternative ways to aggregate subclasses and prompts.

## G  NOISY AVAILABLE HIERARCHY DETAILS

The ImageNet (Deng et al., 2009) dataset itself includes a rich hierarchical taxonomy, where every class is a leaf node of the hierarchy. In the original BREEDS (Santurkar et al., 2021) work, the authors modify the structure slightly in order to place concepts at semantically-similar levels of granularity at the same depth, and additional restrict the number of subclasses within each of the BREEDS datasets in order to balance the data. Thus, it is possible for each BREEDS dataset to use the dataset with its superclasses and restricted set of subclasses but provide CHiLS with *all* the subclass labels present in the ImageNet hierarchy for each superclass (i.e. all leaf nodes descended from each superclass node). In Table 7, we display a subset of the living17 BREEDS dataset class structure with the original subclasses and the ImageNet subclasses. Observe that in some cases, there are many subclass labels provided to CHiLS than is present in the data.

## H  DATASET DETAILS

| Dataset | Domains |
|---|---|
| BREEDS | ImageNet, ImageNet-Sketch, ImageNetv2, ImageNet-c {Fog-1, Contrast-2, Snow-3, Gaussian Blur-4, Saturate-5} |
| Office31 | Amazon, DSLR, webcam |
| OfficeHome | Clipart, Art, Real World, Product |

Table 6: Domains used for BREEDS, Office31, and OfficeHome.

**CHiLS Across Domain Shifts**  For each of the BREEDS datasets (Santurkar et al., 2021), Office31 (Saenko et al., 2010), and OfficeHome (Venkateswara et al., 2017), all results presented are the average over different domains. The specific domains used are show in Table 6.

**Fruits-360**  For zero-shot classification with CLIP models, Fruits-360 (Mureşan & Oltean, 2018) in its raw form is somewhat ill-formed from a class name perspective, as there are classes only differentiated by a numeric index (e.g. "Apple Golden 1" and "Apple Golden 2") and classes at mixed granularity (e.g. "forest nut" and "hazelnut" are separate classes even though hazelnuts are a type of forest nut). We thus manually rename classes using the structure laid out in Table 9, which results in a 59-way superclass classification problem, with 102 ground-truth subclasses.

**ObjectNet**  The ObjectNet dataset (Barbu et al., 2019) has partial overlap (113 classes) with the ImageNet (Deng et al., 2009) hierarchical class structure. From this subset of ObjectNet, we use the BREEDS hierarchy (Santurkar et al., 2021) to generate a coarse-grained version of ObjectNet that is shown in Table 8. In this 11-way classification task, the true subclasses are the original ObjectNet classes.

| Superclass | Original BREEDS subclasses | All ImageNet subclasses |
|---|---|---|
| salamander | European fire salamander, common newt, eft, spotted salamander | European fire salamander, common newt, eft, spotted salamander, axolotl |
| turtle | loggerhead, leatherback turtle, mud turtle, terrapin | loggerhead, leatherback turtle, mud turtle, terrapin, box turtle |
| lizard | common iguana, American chameleon, agama, frilled lizard | banded gecko, common iguana, American chameleon, whiptail, agama, frilled lizard, alligator lizard, Gila monster, green lizard, African chameleon, Komodo dragon |
| snake | thunder snake, ringneck snake, diamondback, sidewinder | thunder snake, ringneck snake, hognose snake, green snake, king snake, garter snake, water snake, vine snake, night snake, boa constrictor, rock python, Indian cobra, green mamba, sea snake, horned viper, diamondback, sidewinder |
| spider | black and gold garden spider, barn spider, garden spider, black widow | black and gold garden spider, barn spider, garden spider, black widow, tarantula, wolf spider |
| grouse | black grouse, ptarmigan, ruffed grouse, prairie chicken | black grouse, ptarmigan, ruffed grouse, prairie chicken |
| parrot | African grey, macaw, sulphur-crested cockatoo, lorikeet | African grey, macaw, sulphur-crested cockatoo, lorikeet |
| crab | Dungeness crab, rock crab, fiddler crab, king crab | Dungeness crab, rock crab, fiddler crab, king crab |

Table 7: Subset of living17 class hierarchy, showing the difference between the original BREEDS subclasses and the ImageNet subclasses used for the ablation in Section 4.4: Noisy Available Hierarchies.

Table 9: Mapping from original class names to new subclass and superclasses for Fruits-360.

| Original Class | Cleaned Subclass | Cleaned Superclass |
|---|---|---|
| Apple Braeburn | braeburn apple | apple |
| Apple Crimson Snow | crimson snow apple | apple |
| Apple Golden 1 | golden apple | apple |
| Apple Golden 2 | golden apple | apple |
| Apple Golden 3 | golden apple | apple |
| Apple Granny Smith | granny smith apple | apple |
| Apple Pink Lady | pink lady apple | apple |
| Apple Red 1 | red apple | apple |
| Apple Red 2 | red apple | apple |
| Apple Red 3 | red apple | apple |
| Apple Red Delicious | red delicious apple | apple |
| Apple Red Yellow 1 | red yellow apple | apple |
| Apple Red Yellow 2 | red yellow apple | apple |
| Apricot | apricot | apricot |
| Avocado | avocado | avocado |
| Avocado ripe | avocado | avocado |
| Banana | banana | banana |
| Banana Lady Finger | lady finger banana | banana |
| Banana Red | red banana | banana |
| Beetroot | beetroot | beetroot |
| Blueberry | blueberry | blueberry |
| Cactus fruit | cactus fruit | cactus fruit |

| | | |
|---|---|---|
| Cantaloupe 1 | melon | melon |
| Cantaloupe 2 | melon | melon |
| Carambula | star fruit | star fruit |
| Cauliflower | cauliflower | cauliflower |
| Cherry 1 | cherry | cherry |
| Cherry 2 | cherry | cherry |
| Cherry Rainier | rainier cherry | cherry |
| Cherry Wax Black | black cherry | cherry |
| Cherry Wax Red | red cherry | cherry |
| Cherry Wax Yellow | yellow cherry | cherry |
| Chestnut | nut | nut |
| Clementine | orange | orange |
| Cocos | cocos | cocos |
| Corn | corn | corn |
| Corn Husk | corn husk | corn husk |
| Cucumber Ripe | cucumber | cucumber |
| Cucumber Ripe 2 | cucumber | cucumber |
| Dates | date | date |
| Eggplant | eggplant | eggplant |
| Fig | fig | fig |
| Ginger Root | ginger root | ginger root |
| Granadilla | granadilla | passion fruit |
| Grape Blue | blue grape | grape |
| Grape Pink | pink grape | grape |
| Grape White | white grape | grape |
| Grape White 2 | white grape | grape |
| Grape White 3 | white grape | grape |
| Grape White 4 | white grape | grape |
| Grapefruit Pink | pink grapefruit | grapefruit |
| Grapefruit White | white grapefruit | grapefruit |
| Guava | gauva | gauva |
| Hazelnut | nut | nut |
| Huckleberry | huckleberry | huckleberry |
| Kaki | kaki | persimmon |
| Kiwi | kiwi | kiwi |
| Kohlrabi | kohlrabi | kohlrabi |
| Kumquats | kumquat | kumquat |
| Lemon | lemon | lemon |
| Lemon Meyer | meyer lemon | lemon |
| Limes | lime | lime |
| Lychee | lychee | lychee |
| Mandarine | orange | orange |
| Mango | mango | mango |
| Mango Red | red mango | mango |
| Mangostan | mangostan | mangostan |
| Maracuja | maracuja | passion fruit |
| Melon Piel de Sapo | melon | melon |
| Mulberry | mulberry | mulberry |
| Nectarine | nectarine | nectarine |
| Nectarine Flat | flat nectarine | nectarine |
| Nut Forest | forest nut | nut |
| Nut Pecan | pecan nut | nut |
| Onion Red | red onion | onion |
| Onion Red Peeled | red onion | onion |
| Onion White | white onion | onion |
| Orange | orange | orange |
| Papaya | papaya | papaya |
| Passion Fruit | passion fruit | passion fruit |
| Peach | peach | peach |

| | | |
|---|---|---|
| Peach 2 | peach | peach |
| Peach Flat | flat peach | peach |
| Pear | pear | pear |
| Pear 2 | pear | pear |
| Pear Abate | abate pear | pear |
| Pear Forelle | forelle pear | pear |
| Pear Kaiser | kaiser pear | pear |
| Pear Monster | monster pear | pear |
| Pear Red | red pear | pear |
| Pear Stone | stone pear | pear |
| Pear Williams | williams pear | pear |
| Pepino | pepino | pepino |
| Pepper Green | green pepper | pepper |
| Pepper Orange | orange pepper | pepper |
| Pepper Red | red pepper | pepper |
| Pepper Yellow | yellow pepper | pepper |
| Physalis | groundcherry | groundcherry |
| Physalis with Husk | groundcherry | groundcherry |
| Pineapple | pineapple | pineapple |
| Pineapple Mini | mini pineapple | pineapple |
| Pitahaya Red | dragon fruit | dragon fruit |
| Plum | plum | plum |
| Plum 2 | plum | plum |
| Plum 3 | plum | plum |
| Pomegranate | pomegranate | pomegranate |
| Pomelo Sweetie | pomelo | pomelo |
| Potato Red | red potato | potato |
| Potato Red Washed | red potato | potato |
| Potato Sweet | sweet potato | potato |
| Potato White | white potato | potato |
| Quince | quince | quince |
| Rambutan | rambutan | rambutan |
| Raspberry | raspberry | raspberry |
| Redcurrant | redcurrant | redcurrant |
| Salak | salak | snake fruit |
| Strawberry | strawberry | strawberry |
| Strawberry Wedge | strawberry | strawberry |
| Tamarillo | tamarillo | tamarillo |
| Tangelo | tangelo | tangelo |
| Tomato 1 | tomato | tomato |
| Tomato 2 | tomato | tomato |
| Tomato 3 | tomato | tomato |
| Tomato 4 | tomato | tomato |
| Tomato Cherry Red | cherry tomato | tomato |
| Tomato Heart | heart tomato | tomato |
| Tomato Maroon | maroon tomato | tomato |
| Tomato Yellow | yellow tomato | tomato |
| Tomato not Ripened | unripe tomato | tomato |
| Walnut | nut | nut |
| Watermelon | melon | melon |

Table 8: Class Structure for ObjectNet experiments.

| Superclass | Subclasses (Original ObjectNet) |
|---|---|
| garment | {Dress, Jeans, Skirt, Suit jacket, Sweater, Swimming trunks, T-shirt} |
| soft furnishings | {Bath towel, Desk lamp, Dishrag or hand towel, Doormat, Lampshade, Paper towel, Pillow} |
| accessory | {Backpack, Dress shoe (men), Helmet, Necklace, Plastic bag, Running shoe, Sandal, Sock, Sunglasses, Tie, Umbrella, Winter glove} |
| appliance | {Coffee/French press, Fan, Hair dryer, Iron (for clothes), Microwave, Portable heater, Toaster, Vacuum cleaner} |
| equipment | {Cellphone, Computer mouse, Keyboard, Laptop (open), Monitor, Printer, Remote control, Speaker, Still Camera, TV, Tennis racket, Weight (exercise)} |
| furniture | {Bench, Chair} |
| toiletry | {Band Aid, Lipstick} |
| wheeled vehicle | {Basket, Bicycle} |
| cooked food | {Bread loaf} |
| produce | {Banana, Lemon, Orange} |
| beverage | {Drinking Cup} |

