# OpenReview forum: "CHiLS: Zero-Shot Image Classification with Hierarchical Label Sets"
_ICLR.cc/2023/Conference — Submitted to ICLR 2023_

### Official Review · Reviewer_msk3 · 2022-10-12

**Confidence:** 5
**Correctness:** 3
**Technical Novelty And Significance:** 3
**Empirical Novelty And Significance:** 3
**Recommendation:** 5

**Clarity, Quality, Novelty And Reproducibility:**

* The use of GPT-3 in zero-shot open-vocabulary image classification is novel (also appearing in concurrent work as the authors recognize).

* While I have not seen hierarchical classification methods applied in zero-shot image classification, these kinds of techniques are well studied in the literature in more classical ML settings. The authors can improve the manuscript by positioning their hierarchical classification algorithm relative to others. However, the authors do not claim to present hierarchical classification in-and-of-itself as novel.

* I have a minor concern about reproducibility given that the GPT-3 component of the work uses a temperature parameter of 0.7 and hence introduces some randomness into the results. However, the authors release all generated labels and hence the numbers in the paper should be reproducible for the datasets presented.

* Some of the experiments require better explanation and could benefit from appendices giving more experimental details (see W7, W10).

* The introduction and method section are very well written.

**Strength And Weaknesses:**

**Strengths**

*S1.* The work is well motivated.

*S2.* The work seems novel. I have only seen GPT-3 used in zero-shot open-vocab classification in concurrent work and consider this to be novel.

*S3.* The method is general for classification problems and does not require additional training.

*S4.* The ablations test major design decisions and shed light on when a practitioner may or may not want to make similar decisions.

*S5.* Method is simple and easy to implement.

*S6.* The introduction and method section are very well written.

**Weaknesses**

*W1.* Related work could be more comprehensive and the paper could be better situated in the literature. More specifically, here are some suggestions for references. I suggest doing a more comprehensive literature review of recent work.

Transfer learning:
* LiT: tuning text towers while keeping language tower fixed (https://arxiv.org/abs/2111.07991)
* Visual prompt tuning: tuning learnable visual prompt (https://arxiv.org/abs/2203.12119)
* Model Soups: ensembling many fine-tuned CLIP models in weight space (https://arxiv.org/abs/2203.05482)
* Patching CLIP models on downstream tasks while maintaining zero-shot performance on tasks where CLIP is already successful (https://arxiv.org/abs/2208.05592)
* CLIP-CL: similar to Model Patching above (https://arxiv.org/abs/2207.09248)
* ...

Zero-shot prediction:
* CLIP-ViL: showing the limitation of ZS CLIP in VQA-like settings (https://arxiv.org/abs/2107.06383)
* CoW: showing how CLIP can be adapted to do object navigation without additional training (https://arxiv.org/abs/2203.10421)
* ...

Additionally, I suggest adding a section on hierarchical classification.

*W2.* The paper states: "Here, we reweight each set of subclass probabilities by its superclass probability. In this way, we can attempt to avoid the behavior in which CLIP makes an incorrect subclass prediction despite a confident and correct superclass prediction, and thus bias our model to never do worse than the raw superclass predictions."

While this is maybe meant to provide intuition for the method, it reads like a claim that is not supported. I suggest making it clear that this is intuition and not some mathematical/provable property of the algorithm related to "bias".

*W3.* How does the proposed method perform on ImageNet? Without ImageNet evaluation it becomes hard to compare the proposed method to other methods.

*W4.* It seems many of the datasets are chosen because it is easy to construct class hierarchies. However, the method is presented as a general purpose zero-shot image classification improvement over CLIP. To be convinced of this, I would be interested to see results on more datasets. Please consider Cars, DTD, EuroSAT, GTSRB, KITTI, MNIST, RESISC45, SUN397, which Ilharco et al. (https://arxiv.org/abs/2208.05592) use because they are tasks ZS CLIP is known to struggle on. Note: dataloaders for these datasets can be found [here](https://github.com/mlfoundations/patching/tree/main/src/datasets).

*W5.* The main experimental setup has baselines that may not be fair points of comparison. Specifically, not all datasets in the original CLIP paper use the ImageNet prompt templates. For example, CIFAR-10/Food101 use different prompts (see [here](https://github.com/openai/CLIP/blob/e184f608c5d5e58165682f7c332c3a8b4c1545f2/data/prompts.md)). What are the deltas between the 75 prompt set and the set that OpenAI provides? For Food101 specifically, there seems to be a discrepancy of ~2% between the ZS number reported in the manuscript and the L/14-336px number reported in the CLIP paper in Tab 11. For datasets that OpenAI did not evaluate in their original paper, some prompt engineering on a val set or some validation that ImageNet prompts are reasonable seems needed.

*W6.* It seems that the experimental setup is sufficiently different in cases where the hierarchy exists and when it does not for me to compare the performance between col 2 and 3 in Table 1. For example, in the *GPT-3 map* the authors include the superclass label in the label set, while in *Existing Map* they do not. It would be good to standardize the algorithm being compared in the two settings. Another idea is to provide the full results in the appendix (i.e., *GPT-3 map* w/ superclass labels, *GPT-3 map* w/o superclass labels, *Existing map* w/ superclass labels, *Existing map* w/o superclass labels).

*W7.* I am not able to completely understand the experimental setup for “Noisy Available Hierarchies” from the text provided. Consider adding an appendix for that section to give more details on how the subclasses are constructed from the ImageNet hierarchy.

*W8.* How do things look if the label set size is 1 or 100 (i.e., m=1 or m=100)? Doe these more extreme values of m affect the performance? At least m=1 should be worse than zero-shot, which seems like a valuable bases of comparison.

*W9.* Some visualizations of cases where the vanilla zero-shot and CHiLS models disagree could be helpful to provide intuition.

*W10.* I am not clear on the linear class ensemble experiment from reading the text. Is the following correct? For a single class, loop over all subclasses and ImageNet prompt templates computing the CLIP text features. Average all of these features to represent the single class as a feature. Repeat for all classes to get a zero-shot classification head.

*W11.* The method involves expanding the zero-shot head at test time. There is a compute and memory overhead associated with this that may limit the scalability of the method when many classes or subclasses are targeted. For example, for the 1000 ImageNet classes, with m=10, the instantiated head would have 10k classes.

**Minor**

*M1.* The Radford et al. 2021 caption example is “a photo of a {}.” not “a photo of a {}” as presented in the manuscript. The difference is the period at the end of the prompt, which may make a difference in downstream performance.

*M2.* This is a relevant reference when discussing CLIP confidence in the method section: https://arxiv.org/abs/2106.07998

*M3.* For Sec. 4.4 is there some theory as to why weighting is necessary in one case and not in the other?


**Summary Of The Paper:**

The paper provides an algorithm to improve zero-shot open-vocabulary classifiers by better prompting them. The authors propose to (1) create sub-classes for each parent classification category using existing (human-created) or inferred (GPT-3 generated) label hierarchies, (2) perform standard zero-shot classification on the sub-classes, and (3) aggregate the results to associate probabilities with the parent category. The paper provides empirical evaluation of the method on various datasets and additionally ablates key design decisions related to their use of labels, GPT-3, and aggregation strategy.

**Summary Of The Review:**

The paper provides a method to prompt open-vocabulary models for zero-shot image classification, which leverages class hierarchies. It is straightforward to implement and the authors provide some empirical evidence that their method performs significantly better than the baseline, especially when hierarchies are known for a dataset.

However, I currently recommend weak rejection of the manuscript. I am most concerned about the lack of benchmarking on ImageNet (*W3.*), the performance of the method on at least a few more standard classification datasets (*W4.*), the fairness of the zero-shot baselines (*W5.*), and the clarity of the presentation associated with some of the ablations studies (*W7.*, *W10.*).

I am willing to revisit my evaluation during the rebuttal/discussion period.

---

> ### Author Response · Authors · 2022-11-13
> **Reviewer Response (Part 1)**
>
> Thank you for your incredibly detailed response! We have detailed our response to the points you made below:
>
> >**W1. Related work could be more comprehensive and the paper could be better situated in the literature. More specifically, here are some suggestions for references.…Additionally, I suggest adding a section on hierarchical classification.**
>
> The author team thanks the reviewer for this insight. We have modified the related works section to present a more holistic review of the literature, including the citation suggestions recommended. We have additionally added a short section on hierarchical classification discussing references that we feel are the closest to the work we present here.
>
>
> > **W2. The paper states: "Here, we reweight each set of subclass probabilities by its superclass probability. In this way, we can attempt to avoid the behavior …. and thus bias our model to never do worse than the raw superclass predictions." While this is maybe meant to provide intuition for the method, it reads like a claim that is not supported. I suggest making it clear that this is intuition and not some mathematical/provable property of the algorithm related to "bias".**
>
> Thanks for the catch. The authors understand the miscommunication on our part here, and have reworded the above statement to make clear that this is intuition and not a formal property. Additionally, we performed an ablation over possible modifications on the reweighting algorithm itself, in hopes that this too will help shed light on this factor of our proposed method.
>
> > **W3. How does the proposed method perform on ImageNet? Without ImageNet evaluation it becomes hard to compare the proposed method to other methods.**
>
> > **W4. It seems many of the datasets are chosen because it is easy to construct class hierarchies. However, the method is presented as a general purpose zero-shot image classification improvement over CLIP… Please consider Cars, DTD, EuroSAT, GTSRB, KITTI, MNIST, RESISC45, SUN397…**
>
> (W3 and W4 response together) We apologize for the confusion. We realized that we did not effectively communicate the fact that our proposed method is particularly meant for scenarios in which the class labels are ill-formed and at a coarse level of granularity that is difficult for CLIP models, and not as a general purpose zero-shot classification improvement. The reviewer is exactly correct that the datasets are chosen due to their coarse and poorly named labels, as this is exactly the class of datasets that we intend for our method to be used on.
>
> While this restricts the applicability of our method, we believe it would not be a significant issue in practice, as for practitioners it would only take a general sense of the task and class label distribution for a practitioner to assess whether our method could be useful. We have addressed this clarification throughout the paper in order to make this more clear and alleviate any confusion.
>
> Additionally, as per your suggestion, we have included a few of the datasets (namely EuroSAT and RESISC45) as an additional test of our methods effectiveness on tasks ZS CLIP struggles on. These additional results are reflected throughout the paper.
>
> >**W5. The main experimental setup has baselines that may not be fair points of comparison. Specifically, not all datasets in the original CLIP paper use the ImageNet prompt templates… For Food101 specifically, there seems to be a discrepancy of ~2% between the ZS number …**
>
> We sincerely thank the reviewer for this insightful comment and catch.
> - First off, we wanted to bring up the slight note that the baseline discrepancy between our paper and the original CLIP paper for Food-101 was due to the fact that we use the *entire* dataset for zero-shot evaluation, while the CLIP paper only uses the test set (while we are not sure if this is specified in the original paper, we recreated the train-test split to investigate the issue).
> - As per your main comment, we have re-run __every__ experiment in the paper to reflect a more fair comparison with more dataset-specific prompts. Now, for every dataset, we choose either the ImageNet 75 prompts *or* a dataset-specific set of prompt templates (which we reuse from the original CLIP paper in the case of Food101/RESISC45/EuroSAT/CIFAR20 or build ourselves where applicable) that adds “context” to the prompt, whichever performs better on the baseline superclass performance. To make a fair comparison, we additionally add the context to the GPT-3 query whenever the dataset-specific prompt set is used. We have modified the paper to reflect these changes, added additional explanation in Section 4.1, and have included an appendix section to elaborate on this. Notably, **there are no major changes in our results as a consequence of this improved prompt engineering**, as our method continues to perform as it did before optimizing for the best prompt set.

---

> > ### Comment · Reviewer_msk3 · 2022-11-20
> > **Post Rebuttal**
> >
> > Thank you to the authors for your detailed responses and new experiments. However, after reviewing the revised manuscript, I am electing to keep my original score for the following reasons:
> >
> > 1. "ill-formatted" is not clearly defined. More generally the scope of the paper is not a general zero-shot improvement, but rather specialized for cases where class-hierarchies may exist. While this specialization, in and of itself, does not preclude the work to be presented at a conference like ICLR, it does make the contribution more limited. Furthermore, the framing of the paper still does not make the limited scope abundantly clear.
> > 2. The authors state in rebuttal: "the baseline discrepancy between our paper and the original CLIP paper for Food-101 was due to the fact that we use the entire dataset for zero-shot evaluation, while the CLIP paper only uses the test set." I feel that the authors should use the standard zero-shot protocol of evaluating on test sets. The authors also mention that they "recreated the train-test split to investigate the issue." However, they do not specify what the conclusion of their investigation is.

---

> > > ### Author Response · Authors · 2022-11-23
> > > **Post-Rebuttal Response**
> > >
> > > The author team thanks the reviewer for their response, and would like to clarify some things that we did not articulate clearly in our initial rebuttal:
> > >
> > > >**"ill-formatted" is not clearly defined. More generally the scope of the paper is not a general zero-shot improvement, but rather specialized for cases where class-hierarchies may exist. While this specialization, in and of itself, does not preclude the work to be presented at a conference like ICLR, it does make the contribution more limited. Furthermore, the framing of the paper still does not make the limited scope abundantly clear.**
> > >
> > > We apologize for the lack of clarity with regards to both what “ill-formatted” entails and the larger framing of the paper, and have such modified the draft to replace the term “ill-formatted” with “coarsely-defined,” as we think this term better encapsulates our original intention. In Paragraph 2 of the introduction we provide some examples of “coarsely-defined” class labels we would expect our method to be beneficial on (namely “large manmade outdoor things” and “reptiles” from CIFAR20).
> > >
> > > >**The authors state in rebuttal: "the baseline discrepancy between our paper and the original CLIP paper for Food-101 was due to the fact that we use the entire dataset for zero-shot evaluation, while the CLIP paper only uses the test set." I feel that the authors should use the standard zero-shot protocol of evaluating on test sets. The authors also mention that they "recreated the train-test split to investigate the issue." However, they do not specify what the conclusion of their investigation is.**
> > >
> > > We want to clarify that our investigation just entailed running our baseline superclass model (i.e. standard CLIP zero-shot) on the Food-101 test set provided by PyTorch, and observing that the performance matched up more closely with the original CLIP paper. Thus we concluded that the test set was in fact used for Food-101 in their paper.
> > >
> > > The authors were unaware that the standard practice for zero-shot evaluation was to only use the test set, and we are in the process of regenerating our results with this in mind based on your suggestion. Namely, for the Food-101 test set we have the following results (which will be updated in Table 1):
> > >
> > > Superclass: 93.88%
> > >
> > > CHiLS: 93.81%
> > >
> > > We observe that the values align closer to the original CLIP paper, and that the relative performance difference is preserved from our initial results.

---

> ### Author Response · Authors · 2022-11-13
> **Reviewer Response (Part 2)**
>
> (...continued from Part 1)
>
> > **W6. It seems that the experimental setup is sufficiently different in cases where the hierarchy exists and when it does not… For example, in the GPT-3 map the authors include the superclass label in the label set, while in Existing Map they do not…Another idea is to provide the full results in the appendix…**
>
> We apologize here for the lack of clarity, and have updated the appendix to include the full results as you described here. In the main paper, the results presented in Table 1 are presented in this way as they are the “best possible” performance for each map setting controlling for the presence/absence of the superclass label in the label set, and thus can be compared in this fashion. We have kept Table 1 in the same format.  We think that it is important that we show the comparison in terms of best possible performance. This choice can be reasoned about with the following heuristic (which we have also included in Appendix D): When we have access to existing hierarchical information, adding the superclass generally hurts performance because the existing map is correct. When we generate the map with GPT-3, we have no guarantee whether the label set is correct or not (on top of the fact that GPT-3 outputs can be quite noisy), and thus, adding the superclass label helps stabilize the prediction.
>
> > **W7. I am not able to completely understand the experimental setup for “Noisy Available Hierarchies” from the text provided. Consider adding an appendix for that section to give more details on how the subclasses are constructed from the ImageNet hierarchy.**
>
> We apologize for the lack of clarity in explaining this ablation. We have added an additional section in Appendix G in order to explain this setup more thoroughly, as well as an example of the class structure in play for this experiment. Additionally, we will attempt to briefly clarify the experimental setup here.
>
> ImageNet itself includes a rich hierarchical taxonomy from the wordnet taxonomy, where every
> class is a leaf node of the hierarchy. In the original BREEDS paper, the
> authors restrict the number of subclasses within each of the main BREEDS datasets in order to balance the data. So given that for each BREEDS dataset we have the set of $k$ classes with $m$ subclasses for each superclass present in the dataset, but there are $m_i’ \ge m$ subclasses for the $i$th superclass in the full ImageNet hierarchy, we can alternatively provide CHiLS with **all** $m_i’$ subclass labels for each superclass. In this way, we are able to investigate the behavior of CHiLS when the label set for each class includes the correct subclasses **and** subclasses that are out-of-distribution.
>
> > **W8. How do things look if the label set size is 1 or 100 (i.e., m=1 or m=100)? Doe these more extreme values of m affect the performance? At least m=1 should be worse than zero-shot, which seems like a valuable bases of comparison.**
>
> The author team thanks the reviewer for this suggestion. We have since modified our label set size ablation to include both $m=1$ (extreme small) and $m=50$ (extreme large) for each dataset. In general, both the extreme small and extreme large cases tend to show less positive results, though this is not *always* true (see the updated table in the appendix for fine-grained information). $m=1$ itself seems to have very high variability, as in some cases it can maintain some form of accuracy benefit but in others show drastic decreases. Additionally, the median for $m=50$ is noticeable lower than the more moderately sized label sets.
>
> > **W9. Some visualizations of cases where the vanilla zero-shot and CHiLS models disagree could be helpful to provide intuition.**
>
> As per your suggestion, we have added a figure (now Figure 2) to our paper, showing all possible cases of agreement / disagreement between vanilla zero-shot and CHiLS across three (randomly-picked) datasets.
>
> > **W10. I am not clear on the linear class ensemble experiment from reading the text. Is the following correct? For a single class, loop …. classification head.**
>
> We apologize for the lack of clear explanation in this section, and you are exactly correct. We have modified this section in the paper in order to more clearly explain the alternative aggregation methods.

---

> ### Author Response · Authors · 2022-11-13
> **Reviewer Response (Part 3)**
>
> (...continued from Part 2)
>
> > **W11. The method involves expanding the zero-shot head at test time. There is a compute and memory overhead associated with this that may limit the scalability of the method when many classes or subclasses are targeted. For example, for the 1000 ImageNet classes, with m=10, the instantiated head would have 10k classes.**
>
> The author team recognizes this critique especially when there are many superclasses in the original dataset. However, we do not think that such kinds of issues would bottleneck with our method for two reasons:
> - In principle, we are only expanding on the linear head. To our knowledge, scaling up the final linear classification head, even by orders of magnitude, does not tend to actually lead to significant memory overhead (as compared to the full model size which includes backbone).
> - Moreover, our method is aimed to be used only on the class of datasets where some semantic hierarchy could be present (we apologize as this was unclear before rebuttal updates). Thus, we would expect that in the scenario where there are enough superclasses such that generating subclass predictions could pose memory issues (e.g. ~1000+), the superclasses would already be at a fine-enough semantic granularity that it is outside the use case of our method in the first place. We have made this clarification about our method’s class of use cases in the draft.
>
> > **M1. The Radford et al. 2021 caption example is “a photo of a {}.” not “a photo of a {}” as presented in the manuscript. The difference is the period at the end of the prompt, which may make a difference in downstream performance.**
>
> Thank you for the note. We have corrected this fact in the paper, and note that the period was present in our implementation (following Radford et al. 2021), but was accidentally omitted in the paper.
>
> > **M2. This is a relevant reference when discussing CLIP confidence in the method section: https://arxiv.org/abs/2106.07998**
>
> Thank you for the reference. We have now included a citation in our paper.
>
> > **M3. For Sec. 4.4 is there some theory as to why weighting is necessary in one case and not in the other?**
>
> While the authors are currently unaware of a purely theoretical justification for while reweighting is necessary only when the true hierarchy is *not* present, we attribute this difference in behavior to the fact that reweighting multiplicatively combines the superclass and subclass predictions, and thus if subclass performance is sufficient on its own (as is the case when the true hierarchy is available), then combining it with superclass predictions can cause the CLIP model to make more mistakes (as it is more closely following the behavior of the underperforming superclass predictor). We have added an additional note in section 4.4 paragraph 1 to address this.

---

### Official Review · Reviewer_oLxZ · 2022-10-24

**Confidence:** 3
**Correctness:** 3
**Technical Novelty And Significance:** 2
**Empirical Novelty And Significance:** 2
**Recommendation:** 3

**Clarity, Quality, Novelty And Reproducibility:**

In terms of clarity, this paper can be improved further by providing proper introduction of the problem formulation, related works.

The proposed method looks pretty straightforward.
While it is interesting to see the improvement of performance, it is only a modest incremental work over the arts.

**Strength And Weaknesses:**


- The outstanding problem of this paper is that it is of not self-contained. It lacks of proper definition and review of referred models, terminologies, and problem setup. For example, "CLIP", "prompt", "open vocabulary models".

- The idea of dividing a super class into a set of subclass is not new, and therefore the contribution is not significant enough.

- The sensitivity of the proposed model on the different levels of granularity of the class hierarchy is unclear.

- Lacks of proper comparisons with related works in line of zero-shot image classification.

**Summary Of The Paper:**

This paper studies zero-shot image classification with class hierarchy.
A Classification with Hierarchical Label sets (CHiLS) model is proposed to improved classical CLIP model.
This model leverages predefined subclasses, and perform CLIP on them first to obtain a set of class embeddings.
These subclass embeddings are aggregated together to form the embedding of class of interests that facilitate zero-shot image classification.



**Summary Of The Review:**

The paper improves the previous arts CLIP by leveraging class hierarchy, which can be seen as a modest incremental work.
Unfortunately, the reviewer does not find it is novel enough.
Besides, the writing and empirical study can be further improved.

---

> ### Author Response · Authors · 2022-11-13
> **Reviewer Response**
>
> Thank you for your feedback! We have detailed our response to the points you made below:
>
> > **The outstanding problem of this paper is that it is of not self-contained. It lacks of proper definition and review of referred models, terminologies, and problem setup. For example, "CLIP", "prompt", "open vocabulary models".**
>
> We apologize for the lack of clarity in setting up the problem, and have since made multiple edits to the paper. Namely, the literature review and the setup has been considerably lengthened in order to provide a more clear picture. Additionally, we have included Appendix B which is dedicated to introducing some of the concepts you mentioned to present preliminaries for CLIP-like models in the general setting.
>
> >  **The idea of dividing a super class into a set of subclass is not new, and therefore the contribution is not significant enough.**
>
> To our knowledge, we have not seen prior work leveraging hyponyms of a given set of classes to improve zero-shot image classification in this fashion. Please let us know if we are missing on important related work.
>
> We want to note that we *do not* claim that we propose the idea of dividing a super class into a set of subclass. Instead, we want to emphasize that given this fact, we show how we can leverage it into the open vocabulary zero-shot inference pipeline and actually see if it gives improvements on an array of benchmarks. There are crucial design decisions to actually see improvements over superclass (naive baseline) with CHiLS.
>
> The main contribution of our work is to propose a simple baseline and perform extensive experiments and careful ablations over its design choices which we believe is itself a worthy contribution.
>
> > **The sensitivity of the proposed model on the different levels of granularity of the class hierarchy is unclear.**
>
> The authors apologize for the lack of clarity here, and will attempt to clarify any confusion here. In our experiments on label set size, we have now included set sizes m=1 and m=50 as per your suggestions. We observe that there is very little consistent change in the quality of our method based on varying this quantity, though we do note that performance tends to be more unstable at these new extreme values (which is to be expected). Additionally, our results on the BREEDs datasets also implicitly test the sensitivity of our method on varying granularity, as living17 and nonliving26 are both finer-grained partial subsets of entity13 and entity30, as all the BREEDs datasets come from the ImageNet hierarchy.
>
>
> > **Lacks of proper comparisons with related works in line of zero-shot image classification.**
>
> To the best of our knowledge, we are not aware of any paper that leverages label hierarchy information for zero-shot image classification in any way that can allow us to make a fair comparison against our method.
>
> However, we do compare our method against a number of different and simpler algorithms that utilize label hierarchies. For instance, in 4.4 section 1 we compare our method with the simpler algorithm of only using subclass predictions, with no reweighting, and show that our method outperforms this baseline when the true hierarchy is not known, and in 4.4 section 6, we show that our method outperforms different ensembling mechanisms. We have added an additional comparison to different reweighting mechanisms (now 4.4 section 2) wherein we show our method outperforms other possible algorithms.
>
> We hope to hear back from the reviewer in case they know of a method that we should compare with. We will be happy to run additional experiments.
>
> > **The paper improves the previous arts CLIP by leveraging class hierarchy, which can be seen as a modest incremental work. Unfortunately, the reviewer does not find it is novel enough. Besides, the writing and empirical study can be further improved.**
>
> Based on your and suggestions from other reviewers, we have significantly improved the clarity of writing and empirical investigation throughout the paper. Changes include:
> - Multiple modifications throughout the paper to address clarity issues brought up by your and the other reviewers, and additions to the appendix explaining some of these issues more in depth
> - An extended and more thorough review of the current literature
> - Standardizing experimental results for each dataset to control for the effect of optimal prompt templates
> - Additional ablations on the choice of reweighting algorithm
> - Further experiments on the effect of the label set size, expanding into the extreme edge cases of this scenario.
> - A set of example images to visually display the behavior differences between the baseline and our method

---

### Official Review · Reviewer_ohpn · 2022-10-25

**Confidence:** 4
**Correctness:** 4
**Technical Novelty And Significance:** 3
**Empirical Novelty And Significance:** 2
**Recommendation:** 6

**Clarity, Quality, Novelty And Reproducibility:**

The paper is clearly written and proposes a simple, yet conceptually novel approach to utilize label hierarchy.
The authors provide code and most implementation details seem to be available.

**Strength And Weaknesses:**

Strengths:

- (S1) A practical and easy-to-implement/utilize method
- (S2) In the case of existing class hierarchy, the experiments indicate a very significant improvement over not relying on that class hierarchy (although see W2)
- (S2) The experiments cover w/ and w/o hierarchical explicit information (although see W1)


Weaknesses:
- (W1) The paper doesn’t clearly show whether the proposed approach is valid/useful only on datasets where some hierarchical structure between classes exist (regardless whether explicitly present or not) or is something that would work on any generic problem/datasets. The authors do address in their experiments two scenarios: one with hierarchical information provided explicitly and another with a hierarchy generated from a GPT model. However, a good class structure/hierarchy may or may not exist even if it is not explicitly provided or utilized. Taking ImageNet as an example - there are many animal/plant species with a very deep hierarchy, many closely related classes on one hand, but on the other hand, some classes have a very shallow hierarchy and few only loosely related classes (I would suppose maybe classes like “cliff” or “traffic sign”?). The authors however seem to choose datasets where one could expect some hierarchy to exist, even if not explicitly present. How would the method behave on datasets like e.g. StanfordCars where all classes are somewhat similar and it’s unclear if some meaningful hierarchy exists, or even the whole ImageNet where maybe there are groups of classes with a nice hierarchy/structure and groups of classes where such hierarchy might not exist?
- (W2) The authors present results for their method but no comparisons to any alternative approaches. One would expect at least some simple baselines that utilize the hierarchical structure of labels.
- (W3) On many datasets the improvement in accuracy is very significant, but on some other, like “living17” or “fruits-360” there’s a relatively much smaller improvement. This is not explained/discussed by the authors - is it something related to some properties of the hierarchies?

**Summary Of The Paper:**

The paper proposes a method to improve predictions of the CLIP model by utilizing a potential structure/hierarchy among classes. Instead of generating predictions on the target classes only, the authors propose to generate predictions on a set of all target labels’ subclasses (which are more granular) and use predictions in subclass space to decide which target class (superclass) to output.
For finding subclasses the authors use either use an existing hierarchy of labels or generate them by prompting a GPT-3 model.

**Summary Of The Review:**

Not so clear how general or practically useful the method is: how specific the results are to datasets/problems with some class structure (whether explicit or implicit). The most significant improvements are when utilizing an existing hierarchical structure of labels but it’s unclear how much of that improvement is specific to the proposed method as no alternative approaches or simple baselines utilizing such information are considered.

**EDIT: Updated the score to lean towards acceptance. The authors' comments are generally convincing and the updated version of the paper makes the scope of the contributions clearer.
Also, the additional results provide more insight into the model's performance and in which scenarios it is expected to work well.**

---

> ### Author Response · Authors · 2022-11-13
> **Reviewer Response**
>
> Thank you for your detailed response! We have written our response to the points you made below:
>
> > **(W1) The paper doesn’t clearly show whether the proposed approach is valid/useful only on datasets where some hierarchical structure between classes exist … or is something that would work on any generic problem/datasets. The authors do address in their experiments two scenarios…. However, a good class structure/hierarchy may or may not exist even if it is not explicitly provided or utilized … How would the method behave on datasets like e.g. StanfordCars …  or even the whole ImageNet ….**
>
> We apologize for the confusion. We realized that we did not effectively communicate the fact that our proposed method is particularly meant for scenarios in which the class labels are ill-formed and at a coarse level of granularity that is difficult for CLIP models, and not as a general purpose zero-shot classification improvement. While this restricts the applicability of our method, we believe that this restriction would not be a significant issue in practice, as in most use-cases practitioners could assess whether the task at hand contains any latent hierarchical structure and decide to use our method depending on that initial assessment. We have addressed this clarification throughout the paper in order to make this more clear and alleviate any confusion.
>
> With regards to the choice of datasets, the author team would like to note that of the datasets chosen, a few of them have weak or possibly non-existent class structures themselves. Notably, Food-101 is of the latter case described by the reviewer, in that some classes seem to have nice structure (e.g. hamburger), but most are very specific and likely have no reasonable hyponyms (e.g. spaghetti bolognese). In this scenario, we do show that CHiLS perform slightly worse than the baseline, thus showing that CHiLS retains most of the baseline performance in cases with non-existent class structure. Office31 and OfficeHome are both also examples of datasets with very little semantic hierarchy present (e.g. classes like ‘desk lamp’, ‘eraser’, and ‘printer’), and in these cases we see improvement and only a slight performance drop (respectively) with our method. We have made edits to the paper based on your comment in order to highlight the variation in semantic granularity among datasets used.
>
> Per your suggestion, we have additionally added two more datasets, EuroSAT and RESISC45, which are both datasets that CLIP tends to struggle on and do not have a clear semantic hierarchy (as they are both aerial image datasets and vary a lot on the semantic granularity of each class).
>
> > **(W2) The authors present results for their method but no comparisons to any alternative approaches. One would expect at least some simple baselines that utilize the hierarchical structure of labels.**
>
> To the best of our knowledge, we are not aware of any published baseline that can leverage label hierarchy information for zero-shot classification (e.g. CLIP).
>
> - Given the simplicity, we want to pitch our method as a baseline method for improving CLIP models in certain scenarios that is orthogonal to the literature on prompt engineering.
> - Within our paper, we do compare our method against a number of different and simpler algorithms that utilize label hierarchies. For instance, in 4.4 section 1 we compare our method with the simpler algorithm of only using subclass predictions, with no reweighting, and show that our method outperforms this baseline when the true hierarchy is not known, and in 4.4 section 6, we show that our method outperforms different ensembling mechanisms. We have added an additional comparison to different reweighting mechanisms (now 4.4 section 2) wherein we show our method outperforms other possible algorithms.
> - Moreover, we would be happy to run additional baselines if the reviewer can point us to specific paper suggestions.
>
> > **(W3) On many datasets the improvement in accuracy is very significant, but on some other, like “living17” or “fruits-360” there’s a relatively much smaller improvement. This is not explained/discussed by the authors - is it something related to some properties of the hierarchies?**
>
> We thank the reviewer for this question, and will attempt to answer it here to the best of our ability. In our analysis, the smaller improvement in some of our datasets may be for one of two reasons. In the case of living17, the baseline performance is already quite good (90+), and thus, it is much harder for our method to improve upon the already well-performing baseline. In the situation of fruits360, we think that this is an artifact of the hierarchy and dataset itself, as specifying the kind of a type of fruit (e.g. a “red delicious apple” for “apple”) may not significantly improve separation from a *visual* context, as internal visual semantic variation within each subset of fruits may not vary that much.

---

### Official Review · Reviewer_VtPj · 2022-10-26

**Confidence:** 3
**Correctness:** 3
**Technical Novelty And Significance:** 3
**Empirical Novelty And Significance:** 3
**Recommendation:** 5

**Clarity, Quality, Novelty And Reproducibility:**

The organization of the paper and the overall writing is clear. And one should be able to reproduce with the firm understanding of the CLIP model.

**Strength And Weaknesses:**

Strength:
The paper is generally well-written with a clear motivation and decent performance gain.

Weakness:
The main concern would be the proposed model is essentially an ensemble of sub-class based CLIP models, though authors address this concern via comparisons against linear average approach, the novelty is still somewhat limited.

**Summary Of The Paper:**

In this paper, authors propose to utilize the label hierarchies to boost the performance of CLIP for zero shot classification. The main steps include the generation of the subclasses for each class by either using the GT label hierarchies or by querying GPT-3, then conduct the CLIP  via these sub-classes, and finally map the sub-classes back to their parents. The performance gain has been observed on the benchmark datasets.

**Summary Of The Review:**

Overall, the paper is well-written, and easy to follow, yet the main concern as expressed in the weakness part is the novelty, as it resembles a lot with an ensemble model.

Moreover, if the hierarchical structure does not explicitly exist in the label vocabulary, how would the proposed model handle it, namely, you cannot rely on the ready-to-go GT structure or query it via the GPT model. It is somewhat unfair if the compared models can also utilize these structures to boost their performance. Authors are suggested to clarify their proposed model in a more general sense.

---

> ### Author Response · Authors · 2022-11-13
> **Reviewer Response**
>
> Thank you for your detailed response! We have detailed our response to the points you made below:
>
> > **Weakness: The main concern would be the proposed model is essentially an ensemble of sub-class based CLIP models, though authors address this concern via comparisons against linear average approach, the novelty is still somewhat limited.**
>
> There are some crucial differences between naive ensembling and CHiLS that we want to highlight. In the naive ensembling approach, subclass text embeddings are averaged together to obtain a linear head for the corresponding superclass. However, in CHiLS, the subclasses across all superclasses are treated as __separate__, i.e., we perform the standard zero-shot CLIP procedure as though these subclasses were the labels of interest, and then map the predicted subclass back to its parent to produce the final prediction.
>
> We experimentally compare the naive ensembling and CHiLS and observe that the design decisions in CHiLS methodology are crucial to its superior performance (i.e., in the GPT-3 map case doing naive ensembling (72.40%) barely even recovers superclass performance (72.28%) on average).
>
> We note that to our knowledge our method is the first to leverage hierarchical class information in this fashion for zero-shot inference with CLIP models.
>
> > **Moreover, if the hierarchical structure does not explicitly exist in the label vocabulary, how would … cannot rely on the ready-to-go GT structure or query it via the GPT model. It is somewhat unfair if the compared models can also utilize these structures to boost their performance. Authors are suggested to clarify their proposed model in a more general sense.**
>
> The author team apologizes for this miscommunication and want to make clear that our method is not a general purpose method, but is aimed to improve performance on the class of datasets where the labels are at a relatively coarse level of semantic granularity. While this restricts the applicability of our method, we believe it would not be a significant issue in practice, as in most use-cases practitioners could assess whether the task at hand contains any latent hierarchical structure and decide to use our method depending on that initial assessment. We have addressed this clarification throughout the paper in order to make exposition of the paper clear and alleviate any confusion.

---

### Author Response · Authors · 2022-11-13
**Overall Response**

The author team sincerely thanks all the reviewers for their time and effort they put into their detailed responses. Given that the overall scores lean negative, we wanted to address some misconceptions about the paper and improve the clarity of understanding in multiple parts. Additionally, per reviewers feedback, we have made several changes to the paper that we believe strengthens its contributions. Next, we provide an overview of the major changes and answer some common concerns.

- A common issue brought up by reviewers (VtPj, ohpn, and msk3) was that the initial draft of the paper presented CHiLS as a general purpose method for zero-shot classification but only focused on datasets where we could reasonably expect a hierarchy to exist.
  - This was a miscommunication not properly addressed in the original draft. We will clarify it here: **We do not intend to present our method as a general purpose improvement for zero-shot classification, but rather an improvement that can be made when the data is at a sufficiently coarse level of semantic granularity.** As our method is grounded in the idea of leveraging latent hierarchical information (whether given or not) to improve zero-shot classification, we do not expect our method to be particularly useful when the original superclass labels have no further levels of granularity. We have made several edits throughout the paper to make the exposition on this part clearer.
  - While this is a limitation of our method and restricts its uses cases, we believe that this limitation will not hinder the applicability of our method, as practitioners would a priori know if their task contains any latent semantic hierarchy and thus choose to use or not use our method based on this knowledge.
- Based on remarks about the lack of baseline comparisons, especially with regards to the reweighting mechanism, we added an additional ablation over varying alternative methods for reconciling subclass and superclass predictions together.
- Thanks to reviewer msk3’s point that comparisons across datasets would be fairer if we tried basic prompt engineering of the prompt templates (which was an issue we were not aware of), we have since rerun **every** experiment in the paper to reflect the fact that for each dataset. We now compare the 75 ImageNet templates with a dataset-specific prompt template set, and use whichever set achieves better baseline performance for all experiments.
- On reviewers ohpn's and msk3’s suggestion about adding more datasets, we have added two new datasets to our analysis, EuroSAT and RESISC45, which are both datasets CLIP is known to struggle on. In both new datasets, we see modest improvements with using our method over the baseline.
- We have added a new figure (Figure 2) that shows some actual examples of how our method compares to the baseline across 3 different datasets and all agreement scenarios.

---

### Decision · Program_Chairs · 2023-01-20

**Decision:**

Reject

**Justification For Why Not Higher Score:**

The paper is missing analyses that are important for readers to understand when they should use the proposed method.

**Justification For Why Not Lower Score:**

N/A

**Metareview: Summary, Strengths And Weaknesses:**

The paper proposes an approach to improve the performance of CLIP for zero-shot image classification by utilizing class hierarchies (or using GPT-3 to access such hierarchies) to make predictions for all subclasses of a class, and then mapping the subclass predictions back to the parent class to get a final prediction.

The key advantage of the approach is that it is a simple adaptation of existing methods that can lead to performance improvements on some image-recognition problems. The main problem with the paper is that it remains unclear which image-recognition problems: in what scenarios does the approach help and in which scenarios does it not help (or even hurt)?

The initial version of the paper somewhat overstated how generally applicable the method is. While that issue appears to have been addressed in the revised paper, the underlying question still remains. Answering this question likely requires more in-depth analysis, for example, on the ImageNet dataset. (One could imagine doing experiments on different versions of the ImageNet dataset in which target classes are chosen at different levels from the WordNet hierarchy as a way to facilitate more systematic analysis.) Doing such analyses is important for an ICLR audience to understand the applicability of the proposed method.